# Study of Different Carbon Bond 6 (CB6) Mechanisms by Using a Concentration Sensitivity Analysis

Le Cao, Simeng Li, and Luhang Sun

Key Laboratory for Aerosol-Cloud-Precipitation of China Meteorological Administration, Nanjing University of Information Science and Technology, Nanjing, China

**Correspondence:** L. Cao
(le.cao@nuist.edu.cn)

**Abstract.** Since the year 2010, different versions of the Carbon Bond 6 (CB6) mechanism have been developed, to accurately estimate the contribution to the air pollution by the chemistry. In order to better understand the differences in simulation results brought about by the modifications between different versions of the CB6 mechanism, in the present study, we investigated the behavior of three different CB6 mechanisms (CB6r1, CB6r2 and CB6r3) in simulating ozone ($O_3$), nitrogen oxides ($NO_x$) and formaldehyde (HCHO) under two different emission conditions, by applying a concentration sensitivity analysis in a box model. The results show that when the surface emission is weak, the $O_3$ level predicted by CB6r1 is approximately 7 ppb higher than that predicted by CB6r2 and CB6r3, specifically due to the change in the sink of acyl peroxy radicals with high order carbons (i.e. species CXO3) in the mechanism and the difference in the ozone dependence on the isoprene emission. In contrast, although CB6r1 estimates higher values of $NO_x$ and HCHO than the other two mechanisms at an early stage of the simulation, the levels of $NO_x$ and HCHO estimated by these three CB6 mechanisms at the end of the 7-day simulation are mostly similar, when the surface emission is weak. After the increase of the surface emission, the simulated profiles of $O_3$, $NO_x$ and HCHO obtained by CB6r2 and CB6r3 were found nearly the same during the simulation period, but CB6r1 tends to estimate substantially higher values than CB6r2 and CB6r3. The deviation between the $O_3$ levels provided by CB6r1 and the other two CB6 mechanisms (i.e. CB6r2 and CB6r3) was found enlarged compared with the weak-emission scenario, because of the weaker dependence of ozone on the emission of isoprene in CB6r1 than those in CB6r2 and CB6r3 in this scenario. Moreover, HCHO predicted by CB6r1 was found larger than that predicted by CB6r2 and CB6r3, which is caused by an enhanced dependence of HCHO on the emission of isoprene in CB6r1. Regarding to $NO_x$, it was found that CB6r1 gives a higher value than the other two mechanisms, which is caused by the relatively stronger connection between the $NO_x$ prediction and the release of NO and $NO_2$ in CB6r1, due to the change in the product of the reaction between isoprene and $NO_3$ in CB6r1. Consequently, more emitted $NO_x$ is involved in the reaction system denoted by CB6r1, which enables a following $NO_x$ formation and thus a higher $NO_x$ prediction of CB6r1.

## 1 Introduction

Air pollution occurs when the concentration of particles or gases in the atmosphere is high, which brings a harmful effect to the human beings and the environment of the earth. It was estimated that in 2007, approximately 3.45 million people were killed

worldwide due to the air pollution (Zhang et al., 2017a). Thus, it is needed to investigate the physicochemical properties of the air pollution so that the formulation of the control strategy by the government can be guided.

Atmospheric transport model (ATM) is an efficient tool for revealing factors dominating the air pollution. Usually the ATM includes a variety of processes that are responsible for the concentration change of pollutants in the atmosphere, such as the production/consumption by the local chemistry, horizontal advection and vertical convection. By using ATMs, the contribution
to the concentration change of the pollutants by each process can be numerically estimated.

Gas-phase chemical reaction mechanism is an essential part of the ATM. It can transform the emissions and the chemical reactions occurring in the atmosphere into the corresponding change of the species, which enables following computations of the ATM. To the present, several atmospheric gas-phase chemical reaction mechanisms have been proposed, such as the detailed Master Chemical Mechanism (MCM) (Jenkin et al., 1997, 2003, 2012, 2015; Saunders et al., 2003; Bloss et al., 2005)
and the global chemical transport model MOZART (Model for Ozone and Related chemical Tracers) mechanism (Emmons et al., 2010). Among these chemical mechanisms, condensed mechanisms such as Carbon Bond Mechanisms (Gery et al., 1989; Zaveri and Peters, 1999; Yarwood et al., 2005, 2010) and SAPRC mechanisms (Carter, 2000a,b, 2010) are widely applied in ATMs due to their relatively small size and adequate accuracy. In these condensed mechanisms, different techniques are used to lump volatile organic compounds (VOCs) into functional groups while the treatment of the inorganic chemistry is mostly
similar.

Carbon Bond Mechanism (CBM) is a kind of condensed mechanism, which lumps VOCs by chemical moiety (Gery et al., 1989; Stockwell et al., 2019). In CBM mechanisms, the carbon bond is treated as a reaction unit, and the carbon bonds with the same bonding state are treated as a group, while the exact location of the carbon bonds in the molecule is neglected. CBM is conveniently implemented in the ATMs because of its small size and high accuracy in predicting the concentration
change of the pollutants. However, due to the lumping technique, biases are inevitably brought into computations. Thus, many updates were made to the CBM mechanism to reduce biases, such as adding an explicit representation of species with the same molecular type (e.g. species ALDX for higher-order aldehydes).

As mentioned above, CBM mechanism has been updated for several generations. In 1989, CB-IV was proposed by Gery et al. (1989), and it was then widely used in many air quality models such as WRF-Chem (Grell et al., 2005) and CMAQ (Byun
and Schere, 2006). In 2005, based on the CB-IV mechanism, Yarwood et al. (2005) released CB05 by explicitly adding higher order aldehydes (ALDX) and internal olefins (IOLE) into the mechanism. A large amount of smog chamber experiments were also used to validate CB05, and it was reported that CB05 behaves better than CB-IV against the chamber data (Yarwood et al., 2005). Later on, an update to CB05 was made by Whitten et al. (2010) by combining a new toluene mechanism with CB05, namely CB05TU mechanism. It was proved that the CB05TU mechanism improves upon the CB05 mechanism in simulating
toluene related reactions (Whitten et al., 2010).

The latest version of the CBM mechanism is CB6 (Yarwood et al., 2010), because it is the 6-th generation of this mechanism family, and it was released to deal with the tightening of the ozone standard in the US. Long-lived and relatively abundant organic compounds formed by peroxy radical reactions ($RO_2$-$RO_2$) are taken into account in CB6. Moreover, the isoprene chemistry and the aromatic chemistry are extensively revised, to improve the modeling of the formation of secondary organic

aerosols (SOAs). It was shown that CB6 performs better in simulating the maximum value of ozone as well as the ozone formation rate compared with the CB05 mechanism (Yarwood et al., 2010). From then on, several updates were made to CB6 so that currently there are four versions of CB6 available, i.e. CB6r1, CB6r2, CB6r3 and CB6r4. In CB6r1, the mechanism previously proposed by Yarwood et al. (2010) was revised again (Yarwood et al., 2012), and several reactions and products were corrected. New experimental data (EUPHORE experiments) were also adopted to validate the CB6r1 mechanism (Yarwood et al., 2012). After that, in 2013, Ruiz Hildebrandt and Yarwood (2013) included the interactions between organic aerosols and total reactive nitrogen ($NO_y$) in the mechanism, and then gave the CB6r2 mechanism. In CB6r2, organic nitrates were divided into two groups, simple alkyl nitrates (NTR1) that remain in the gas phase and multi-functional nitrates (NTR2) that can partition into organic aerosols. Because of the inclusion of the multi-functional aerosol nitrates (i.e. NTR2), lower recycling efficiency of nitrogen oxides ($NO_x$) from nitrates is acquired using CB6r2, leading to a lower ozone production relative to CB6r1 (Ruiz Hildebrandt and Yarwood, 2013). The third version of CB6 is CB6r3 (Emery et al., 2015), which was developed to account for the influence of the low temperature on the formation of organic nitrates. It aims at modeling the winter ozone formation event in Uinta Basin in the US under cold conditions, and it was found that the inclusion of the temperature dependence in CB6r3 would cause an ozone reduction in winter environments, due to an enhanced formation of organic nitrates (Emery et al., 2015). The latest version of the CB6 mechanism is CB6r4 (Emery et al., 2016), which was designed by combing CB6r3 with a 16-reaction skeletal iodine mechanism, to consider the ozone depletion by the iodine chemistry. It was found that CB6r2 and CB6r4 perform similarly in simulating ozone across the continental US, but CB6r4 tends to predict a lower ozone than CB6r2, possibly due to the depletion of ozone by the iodine chemistry in the marine boundary layer (Emery et al., 2016). Currently, the CB6r3 mechanism is available in the latest version of the CMAQ model (Community Multiscale Air Quality model, available at: www.epa.gov/cmaq) (Byun and Schere, 2006), while CB6r2 and CB6r4 are both included in the CAMx model (Comprehensive Air Quality Model with Extensions, available at: www.camx.com) (ENVIRON, 2015; Ramboll Environment and Health, 2020).

Many investigations have been made using the CB6 mechanisms. To name a few, Luecken et al. (2019) used CB6r3 to simulate ozone, oxidized nitrogen ($NO_y$) and hazardous air pollutants (HAPs) across the continental US. In their study, a comparison between CB6r3, CB05TU, and CB05 as well as the observational data was performed. It was shown that these chemical mechanisms behave similarly for the ozone prediction, and CB6r3 performs the best in simulating the vertical distribution of peroxyacyl nitrates. Marvin et al. (2017) used five chemical mechanisms including CB6r2 to evaluate the impact of the isoprene chemistry on the simulation of formaldehyde (HCHO) in the summertime southeast US. They also suggested a set of modifications to CB6r2 that can improve the comparison of the modeled HCHO to observations. Zhang et al. (2017b) used the CAMx model (ENVIRON, 2015; Ramboll Environment and Health, 2020) with the implementation of the CB6r2 mechanism to estimate the biogenic isoprene emissions in US by using two different emission models, BEIS (Pierce et al., 1998; Bash et al., 2016) and MEGAN (Guenther et al., 2006, 2012), and they found that the MEGAN model predicts more isoprene emissions than the BEIS model. Recently, by implementing seven different chemical mechanisms including CB6r3 into a box model constrained by the observational data, Derwent (2017) investigated the responses of the ozone production rate and the mixing ratio of hydroxyl radicals (i.e. OH) to a reduction of $NO_x$ and VOCs in these chemical mechanisms. It

was found that when the constrained values of $NO_x$ and VOCs in the box model are reduced, different mechanisms behave differently, especially in the prediction of OH. Later, Derwent (2020) used the same model to study the response of the OH mixing ratio to the representation of oxidation and degradation of VOCs in thirteen different mechanisms including CB6r3, and Derwent (2020) found that the influence brought about by aromatics such as toluene and o-xylene on the change of OH differs a lot between different chemical mechanisms.

Despite the studies mentioned above, the internal properties of these CB6 mechanisms such as the relationship between the ozone formation and the surface emissions are still not thoroughly investigated and compared. Moreover, the corresponding change brought about by the modifications between different versions of the CB6 mechanism also needs further investigation. Therefore, in this study, we performed a concentration sensitivity analysis on different versions of the CB6 mechanism (CB6r1, CB6r2, and CB6r3) to see the dependence of the formation of ozone ($O_3$), nitrogen oxides ($NO_x$, $NO+NO_2$) and formaldehyde (HCHO) on each reaction of the mechanism as well as the surface emission and dry deposition. By doing that, we were able to figure out reasons causing the deviations between the results obtained by using different CB6 mechanisms. The factors dominating the formation and consumption of the focused species ($O_3$, $NO_x$ and HCHO) in these mechanisms can also be revealed.

The structure of this paper is as follows. In Sect. 2, CB6 mechanisms used in this study are introduced, and the method used to analyze the mechanism as well as the governing equations are also described. Sect. 3 gives the results of the concentration sensitivity analysis and the related discussions. In Sect. 4, major conclusions achieved in this study are summarized. Future work is also prospected in this section.

## 2   Description of the Mechanisms and the Numerical Method

In the present study, we first implemented different versions of the CB6 mechanism (i.e. CB6r1, CB6r2, and CB6r3) into a box model, KINAL (Turányi, 1990a), to simulate the temporal evolution of $O_3$, $NO_x$ and HCHO under two different emission conditions. The surface emission intensity implemented in the model was assumed weak at first, which represents an emission condition in rural regions (Saylor and Ford, 1995; Sandu et al., 1997). By doing that, chemical reactions playing an important role in the change of the focused species can be indicated. Then, sensitivities of the focused species ($O_3$, $NO_x$ and HCHO) to each reaction of the mechanisms were computed, to reveal the influence brought about by the modifications between these CB6 mechanisms. Later, the surface emission was increased in the model, and a same procedure was performed on these mechanisms again, so that the behavior of these CB6 mechanisms under a typical heavily polluted condition in urban regions can be investigated.

The CB6 mechanisms studied in this paper contain approximately 80 chemical species and 220 reactions. The CB6r1 version contains 80 species and 222 reactions, and the CB6r2 version contains 81 species and 215 reactions. The CB6r3 version has 82 species and 221 reactions, including reactions accounting for the temperature dependence of the alkyl nitrate formation. Complete listings of all the reactions of these mechanisms are given in Tab. A1 of the appendix. The updates in CB6r2 and CB6r3 compared with their previous version are also marked in Tab. A1. Compared with CB6r1, CB6r2 divides the organic

nitrates generated from alkanes, olefins, aromatics and oxygenated VOCs (i.e. the species named NTR in CB6r1) into two groups, NTR1 that exists exclusively in the gas phase and NTR2 that can partition into organic aerosols. As a result of this
speciation, in CB6r2, the organic nitrates, NTR1 and NTR2, undergo the following reactions:

$$\text{NTR1} + h\nu \rightarrow \text{NO}_2, \tag{R1}$$

$$\text{NTR1} + \text{OH} \rightarrow \text{NTR2}, \tag{R2}$$

$$\text{NTR2} + \text{H}_2\text{O}_{(\text{aerosol})} \rightarrow \text{HNO}_3. \tag{R3}$$

Reaction (R1) denotes the photolysis of NTR1, which enables a recycling of $NO_x$ and a following ozone formation enhancement. Reaction (R2) represents an addition reaction leading to the conversion from NTR1 to NTR2. Reaction (R3) means that the organic nitrate partitioning within the aerosols undergoes hydrolysis and forms $HNO_3$. Ruiz Hildebrandt and Yarwood (2013) reported that because of this speciation, CB6r2 has a lower recycling efficiency of $NO_x$ from organic nitrates than
CB6r1. The levels of $O_3$ and $NO_x$ predicted by CB6r2 are thus lower than those predicted by CB6r1.

Regarding to CB6r3, it decomposes the formation process of alkyl nitrates from alkanes in CB6r2:

$$\text{PRPA} + \text{OH} \rightarrow 0.71\text{ACET} + 0.26\text{ALDX} + 0.26\text{PAR}$$
$$+ 0.97\text{XO2N} + 0.03\text{XO2N} + \text{RO2}, \tag{R4}$$

$$\text{PAR} + \text{OH} \rightarrow 0.11\text{ALDX} + 0.76\text{ROR} + 0.13\text{XO2N}$$
$$+ 0.11\text{XO2H} + 0.76\text{XO2} + \text{RO2} - 0.11\text{PAR}, \tag{R5}$$

$$\text{XO2N} + \text{NO} \rightarrow 0.5\text{NTR1} + 0.5\text{NTR2}, \tag{R6}$$

into seven reactions:

$$\text{PRPA} + \text{OH} \rightarrow \text{XPRP}, \tag{R7}$$

$$\text{XPRP} \rightarrow \text{XO2N} + \text{RO2}, \tag{R8}$$

$$\text{XPRP} \rightarrow 0.73\text{ACET} + 0.268\text{ALDX} + 0.268\text{PAR}$$
$$+ \text{XO2H} + \text{RO2}, \tag{R9}$$

$$\text{PAR} + \text{OH} \rightarrow \text{XPAR}, \tag{R10}$$

$$\text{XPAR} \rightarrow \text{XO2N}, \tag{R11}$$

$$\text{XPAR} \rightarrow 0.126\text{ALDX} + 0.874\text{ROR} + 0.126\text{XO2H}$$
$$+ 0.874\text{XO2} + \text{RO2} - 0.126\text{PAR}, \tag{R12}$$

$\quad\text{XO2N} + \text{NO} \rightarrow 0.5\text{NTR1} + 0.5\text{NTR2}. \tag{R13}$

By making this modification, the dependence of the alkyl nitrate yield on the pressure and the temperature can be considered in CB6r3, especially under cold conditions. For this purpose, two new operators, XPRP and XPAR, were also added. Under a standard condition (pressure: 1 atm, temperature: 298 K), the formation of the alkyl nitrates (NTR1 and NTR2) in CB6r3 through Reactions (R7)-(R13) is equal to that in CB6r2 through Reactions (R4)-(R6) (Emery et al., 2015).

CB6r4 improves upon CB6r3 by adding a condensed iodine mechanism to consider the iodine-induced ozone destruction (Emery et al., 2016). However, CB6r4 was not investigated in this study, because the halogen chemistry is not the focus of the present study. A comparison between CB6r4 and other CB6 mechanisms in a halogen-rich environment is attributed to a future work.

  We implemented the CB6 mechanisms mentioned above into a box model, KINAL (Turányi, 1990a), to capture the time
variations of $O_3$, $NO_x$ and HCHO by solving Eq. (1):

$$\frac{\mathrm{d}\boldsymbol{c}}{\mathrm{d}t} = \boldsymbol{f}(\boldsymbol{c}, \boldsymbol{k}) + \boldsymbol{E} - \boldsymbol{D}. \tag{1}$$

In Eq. (1), $\boldsymbol{c}$ is a column vector of species concentrations. $\boldsymbol{k}$ is a vector of reaction rate constants and $t$ denotes time. $\boldsymbol{E}$ represents a source term of the local surface emission, and in the present model the surface emission is parameterized as a group of reactions having products and a constant reaction rate but without reactants. $\boldsymbol{D}$ in Eq. (1) is a loss term representing
the dry deposition process of atmospheric constituents, and this process is parameterized in the model as a series of reactions having reactants but without forming any product. KINAL is a box model provided for the analysis of complex reaction systems. Stiff kinetic differential equations can be solved in KINAL, and it was proved that KINAL performs robustly and efficiently (Turányi, 1990a,b; Cao et al., 2014, 2016, 2019). A background air composition (see Tab. 1), adapted from Saylor and Ford (1995) and Sandu et al. (1997), was used as the initial condition of the model. This air composition represents a
heavily polluted atmosphere, in which the background level of $NO_x$ is in the order of 1-100 ppb. Two different scenarios, "weak emission" and "strong emission", were simulated, and the emission intensities belonging to these two scenarios are

listed in Tab. 1. They denote typical emission conditions in rural regions and urban regions, respectively (Saylor and Ford, 1995; Sandu et al., 1997). A 7-day simulation was performed and the simulation starts at noon (12:00) of the first day. The time variations of $O_3$, $NO_x$ and HCHO were recorded every hour during the simulated period.

After obtaining the temporal evolution of $O_3$, $NO_x$ and HCHO, relative concentration sensitivities of these species to different CB6 mechanisms were computed to reveal the dependence of these species on each reaction of the mechanism, surface emissions, and the rate of dry deposition for each atmospheric constituent. The relative concentration sensitivity $\widetilde{S}_{ij}$ can be expressed as

$$\widetilde{S}_{ij} = \frac{\partial \ln c_i}{\partial \ln k_j} = \frac{k_j}{c_i} \frac{\partial c_i}{\partial k_j} = \frac{k_j}{c_i} S_{ij}, \tag{2}$$

which shows the importance of the $j$-th reaction for the concentration change of the $i$-th chemical species. In Eq. (2), $c_i$ is the concentration of the $i$-th chemical species, and $k_j$ denotes the rate constant of the $j$-th reaction. $S_{ij} = \partial c_i / \partial k_j$ is the absolute concentration sensitivity, and the unit of $S_{ij}$ depends on the order of the $j$-th reaction. In order to compare the sensitivity coefficients belonging to different reactions, $S_{ij}$ is normalized by being multiplied with $k_j / c_i$ so that a dimensionless sensitivity coefficient, $\widetilde{S}_{ij}$, is obtained. The relative concentration sensitivity $\widetilde{S}_{ij}$ thus represents the percentage change in the $i$-th species

concentration due to a small perturbation in the rate of the $j$-th reaction. The evaluation of the concentration sensitivity is helpful for discovering the interdependence between the solution of Eq. (1) and input parameters of the model such as the reaction rate constants and the intensity of the surface emission.

     The reaction rate constants of the mechanisms were taken from IUPAC database (Atkinson et al., 2004, 2006, 2007, 2008; Crowley et al., 2010; Ammann et al., 2013) and NASA/JPL database (Sander et al., 2006), and a constant temperature 298 K

was assumed for the calculation of the reaction rates. Photolytic reaction rates were estimated by using TUV (Tropospheric Ultraviolet and Visible) radiation model (Madronich and Flocke, 1997, 1999), assuming a 300 Dobson overhead ozone column and a 1 km measuring height. Data of cross section and quantum yield for each photolyzed species were taken from CMAQ model version 5.3 (Byun and Schere, 2006). When the local time resides between 4:30 (sunrise) and 19:30 (sunset), the photolytic reaction rates vary with the solar zenith angle (SZA), while the photolytic reactions are switched off if the local time

is out of this range. With respect to the dry deposition process, a first-order rate coefficient ($k_d$) indicating the loss caused by dry deposition is calculated using the following equation:

$$k_d = v_d / L, \tag{3}$$

in which $v_d$ denotes the dry deposition velocity, and the values of $v_d$ used in the present study for different atmospheric constituents are given in Tab. 2. $L$ in Eq. (3) is the boundary layer height, and is assumed as 1 km in the model.

In the following section, computational results are presented and discussed.

## 3   Results and Discussions

We first show the temporal evolution of $O_3$, $NO_x$ and HCHO obtained by using CB6r1, CB6r2, and CB6r3 under the given initial condition (see Tab. 1), applying a weak surface emission. The differences between the results using different mechanisms are

also analyzed. Then the concentration sensitivities of the focused species ($O_3$, $NO_x$ and HCHO) to different CB6 mechanisms are displayed, to indicate the internal difference between these mechanisms. Later on, results with the implementation of a strong surface emission are shown. By doing that, the dependence of different CB6 mechanisms on the surface emission under a typical heavily polluted condition in urban regions can be compared and investigated.

## 3.1 Temporal evolution of $O_3$, $NO_x$ and HCHO (weak emission)

Figure 1 shows the temporal profiles of $O_3$, $NO_x$ and HCHO predicted by CB6r1, CB6r2 and CB6r3 in the weak emission scenario. Due to the small intensity of the surface emission in this scenario, the differences between these predictions are able to reflect different capabilities of these mechanisms in converting the initial concentrations into the change of the species. It is seen that under this condition, ozone profiles simulated by these three mechanisms show a notable deviation (see Fig. 1a). In most of the simulated period, the ozone mixing ratio predicted by CB6r1 is higher than those predicted by CB6r2 and CB6r3. Figure 1(a) shows that at the beginning of the simulation (before day 1.5), CB6r1 and CB6r3 behave similarly, while CB6r2 predicts a higher ozone. However, as the reaction proceeds (after day 1.5), CB6r2 starts to predict a lower ozone than CB6r1, and the simulated profile of CB6r2 approaches to that obtained by CB6r3. In contrast, the ozone predicted by CB6r1 becomes higher than those predicted by CB6r2 and CB6r3. During the end of the 7-day simulation, the daily averaged ozone predicted by CB6r1 over the 7-th day is approximately 7 ppb higher than those predicted by CB6r2 and CB6r3, and the ozone levels given by CB6r2 and CB6r3 are almost identical.

For $NO_x$ simulations, Fig. 1(b) shows that at the beginning of the simulation, $NO_x$ declines rapidly from the initial value (7 ppb) to less than 1 ppb, due to the conversion to PAN (peroxyacetyl nitrate) and $HNO_3$. At the end of the simulation, the mixing ratio of $NO_x$ becomes lower than 0.5 ppb. The major nitrogen containing compound during this time period is $HNO_3$, as PAN is thermally decomposed and photolyzed during the daytime. In the comparison of temporal profiles belonging to different mechanisms, CB6r1 consistently estimates a slightly higher $NO_x$ than the other two mechanisms, but the deviation between the estimations of these mechanisms becomes smaller approaching the end of the 7-day simulation. However, it should be noted that the difference in the predicted $NO_x$ using different mechanisms may become larger when the surface emission intensity increases, due to different capability in transforming emissions into the change of the species for each mechanism.

With respect to HCHO predictions, it is seen in Fig. 1(c) that the deviation between the results of these three mechanisms is more pronounced at the start stage of the simulation. During this time period, CB6r1 predicts a much higher HCHO than the other two mechanisms, especially at noon of every day. However, at the end of the simulation, although CB6r1 still estimates a higher HCHO than CB6r2 and CB6r3, the difference becomes smaller, and CB6r2 and CB6r3 give a similar HCHO.

In summary, we found that when the surface emission is weak, the ozone concentration predicted by CB6r1 is mostly higher than that obtained by using CB6r2 or CB6r3. When the end of the simulation approaches, ozone simulated by CB6r1 is approximately 7 ppb larger than those simulated by CB6r2 and CB6r3. In contrast to that, after a 7-day computation, the $NO_x$ and HCHO levels obtained by using these three CB6 mechanisms are more similar. At the beginning of the simulation, CB6r1 gives significantly higher values of $NO_x$ and HCHO than the other two CB6 mechanisms . However, when the end of the simulation comes, the difference tends to disappear.

## 3.2 Concentration sensitivity analysis of different CB6 mechanisms (weak emission)

We then conducted a concentration sensitivity analysis on different CB6 mechanisms under the weak emission condition, and
from these results we were able to identify the relative importance of each reaction in these mechanisms for the change of the
focused species, and discover the reasons causing the deviations between the simulation results of different CB6 mechanisms.

The ozone sensitivity to the CB6r3 mechanism averaged over the last day of the computation is shown in Fig. 2. A positive
sensitivity means that an increase of the reaction rate would accelerate the formation of $O_3$, while a negative value denotes
a decline of $O_3$ when the reaction rate increases. Note that Reactions (R222)-(R234) in Fig. 2(d) represent surface emissions
belonging to different chemical species, and Reactions (R235)-(R243) denote dry depositions for different atmospheric con-
stituents. It is seen from Fig. 2 that apart from the reactions standing for surface emissions and dry depositions, chemical
reactions with large sensitivities mostly possess a reaction number less than 52 (i.e. before Reaction (R52) in the mechanism).
Because Reactions (R1)-(R52) in the mechanism represent the inorganic chemistry while reactions after (R52) are mostly
VOC-involved reactions (see Tab. A1 in the appendix), it demonstrates an important role of the inorganic chemistry in this
simulation, possibly due to the high initial value of $NO_x$ and the weak VOC emissions in this scenario. From the sensitivity
analysis of CB6r3 shown in Fig. 2, we were also able to figure out the most important reactions for the change of ozone, which
can be divided into two groups. The first reaction group includes Reactions (R1) $NO_2 + h\nu \rightarrow NO + O$, (R3) $O_3 + NO \rightarrow NO_2$,
(R25) $HO_2 + NO \rightarrow OH + NO_2$, (R26) $NO_2 + O_3 \rightarrow NO_3$ and (R45) $NO_2 + OH \rightarrow HNO_3$, which are reactions denoting the
inter-conversion of $NO_x$ and the loss of $NO_x$ through the formation of $NO_3$ and $HNO_3$. It demonstrates the significance of reac-
tive nitrogen oxides in determining the final ozone level. The other important reaction group includes (R9) $O_3 + h\nu \rightarrow O(^1D)$,
(R10) $O(^1D) + M \rightarrow O + M$, (R11) $O(^1D) + H_2O \rightarrow 2OH$, and (R13) $O_3 + HO_2 \rightarrow OH$. These reactions represent the ozone
loss due to the formation of hydroxyl radicals (i.e. OH). With respect to the other chemical reactions in the mechanism, their
sensitivities are all smaller than 0.1 (see Figs. 2b, c and d), denoting a minor influence on the change of ozone by these chemical
reactions.

Regarding to the ozone sensitivities to surface emissions (i.e. Reactions (R222)-(R234) in Fig. 2d) and dry depositions (i.e.
Reactions (R235)-(R243) in Fig. 2d), it is seen that although the implemented surface emission in this scenario is weak, the
emission still exerts a strong influence on the change of ozone, indicated by relatively large sensitivity coefficients belonging to
NO and $NO_2$ emissions ($\sim$0.3, see Fig. 2d). The values of the ozone sensitivities to the NO and $NO_2$ emissions are comparable
to those corresponding to chemical reactions denoting the inter-conversion of $NO_x$ (i.e. (R1) and (R3) in Fig. 2a). In contrast
to that, although the emission intensity of isoprene in this scenario is relatively large, the dependence of ozone on the isoprene
emission is minor, reflected by the small ozone sensitivity to the isoprene emission (i.e. Reaction (R232) in Fig. 2d). In addition,
it was also found in Fig. 2(d) that dry deposition is the largest sink of ozone in this weak emission scenario, when CB6r3 is
implemented.

We then computed the ozone sensitivities to the other two mechanisms (i.e. CB6r2 and CB6r1). Because these figures are
similar to Fig. 2, we show these results in the supplementary material of the manuscript (Figs. S1 and S2). The similarity
between these figures also denotes a consistent treatment of the inorganic chemistry and a similar lumping technique of VOCs

in these CB6 mechanisms. By comparing Fig. S1 with Fig. 2, we found that the averaged ozone sensitivity to CB6r2 over the last day is almost identical to the sensitivity to CB6r3, thus leading to a similar ozone prediction by these two mechanisms, which has been shown before (see Fig. 1a). The only major difference between the ozone sensitivities to CB6r2 and CB6r3

is that CB6r3 improves upon CB6r2 by adding Reactions (R217)-(R220) to include the temperature dependence of the alkyl nitrate formation. Therefore, the sensitivities of these reactions are absent for CB6r2, shown in Fig. S1 of the supplements. However, as mentioned above, under a condition of a 298 K temperature and 1 atm pressure, the formation of the alkyl nitrate in CB6r2 and CB6r3 are equivalent. Thus, the addition of these reactions in the CB6r3 mechanism would not significantly affect the predicted ozone in this scenario. However, at a different temperature, this update in CB6r3 may bring a large change

in ozone, indicated by the relatively large sensitivities of ozone to Reactions (R219) $XPAR \rightarrow XO2N + RO2$ and (R220) $XPAR \rightarrow 0.13ALDX + 0.87ROR + 0.13XO2H + 0.87XO2 + RO2 - 0.13PAR$, shown in Fig. 2(d).

We then tried to figure out reactions causing the higher ozone prediction of CB6r1. From a comparison between Fig. S1 and Fig. S2 in the supplementary material, we found two factors heavily responsible for the higher ozone prediction of CB6r1. One is the modification of Reaction (R66) about the sink of CXO3 in CB6r1. CXO3 represents acylperoxy radicals with three

and higher carbons, and is able to oxidize NO and thus form ozone. In CB6r1, the form of Reaction (R66) is as follows, $CXO3 + RO2 \rightarrow CXO3$, while in CB6r2, the form is $CXO3 + RO2 \rightarrow 0.8ALD2 + 0.8XO2H + 0.8RO2$. In CB6r1, the total amount of CXO3 is unaltered through Reaction (R66), thus leading to a negligible ozone sensitivity to (R66). However, the update of Reaction (R66) in CB6r2 causes this reaction to be a major sink of CXO3 in the mechanism. As a result, the ozone significance of Reaction (R66) increases in CB6r2. Moreover, due to this enhanced importance of Reaction (R66), it

was found in CB6r2 that the significance of many other CXO3 related reactions, e.g. (R62) $CXO3 + NO_2 \rightarrow PANX$, (R63) $PANX \rightarrow NO_2 + CXO3$, (R65) $CXO3 + HO_2 \rightarrow 0.41PACD + 0.15AACD + 0.15O_3 + 0.44ALD2 + 0.44XO2H + 0.44RO2 + 0.44OH$, (R67) $CXO3 + CXO3 \rightarrow 2ALD2 + 2XO2H + 2RO2$, and (R110) $ALDX + OH \rightarrow CXO3$ drops, from a moderate value in CB6r1 to a small value in CB6r2. The formation of ozone in CB6r2 is thus getting suppressed due to the additional consumption of CXO3 through Reaction (R66). This finding also denotes an important role of CXO3 in determining ozone

in the CB6 mechanisms. Thus, more attention should be paid to CXO3 related reactions in future mechanism developments. The significance of CXO3 in the mechanism for the conversion of NO to $NO_2$ and the formation of ozone has also been identified by Luecken et al. (2008) in a model study on the behavior of three chemical mechanisms including CB-IV, CB05 and SAPRC99. Aside from the change of Reaction (R66), we also found the ozone sensitivity to the isoprene emission shifts from a small value in CB6r2 and CB6r3 to a moderate positive value in CB6r1 (see Fig. S2(d)). Due to this sensitivity shift,

when CB6r1 is used in the model, an enhanced emission of isoprene in this scenario would result in a significant elevation of the predicted ozone. The reason for this sensitivity shift is attributed to the modification of Reaction (R157) between CB6r1 and the other two mechanisms. In CB6r1, Reaction (R157) is in the form of $ISOP + NO_3 \rightarrow 0.65\,INTR + other\,products$, so that the emitted isoprene is partly converted to INTR (i.e. organic nitrates from isoprene reactions), while in CB6r2 and CB6r3 the product INTR is updated as NTR2. The product INTR in CB6r1 is able to react with OH, forming many organic compounds such as NTR, ALD2 and ALDX that can promote the ozone formation. In contrast, NTR2, the multi-functional nitrate formed through Reaction (R157) in CB6r2 and CB6r3, is then converted to $HNO_3$, which is relatively inactive for the

ozone formation. As a result, the influence caused by the isoprene emission on the change of ozone in CB6r2 and CB6r3 is weaker than that in CB6r1, which is also confirmed by the smaller ozone sensitivity to Reaction (R157) in CB6r2 and CB6r3. Moreover, because of the enhanced ozone dependence on the isoprene emission in CB6r1, the ozone sensitivities to many

reactions that are associated with isoprene and the products of isoprene reactions, e.g. (R156) $ISOP + O_3 \rightarrow products$, (R92) $NTR + h\nu \rightarrow products$, (R106) $ALD2 + OH \rightarrow C2O3$, and (R110) $ALDX + OH \rightarrow CXO3$, in CB6r1 were also found larger than those in CB6r2 and CB6r3 (see Fig. S2). Thus, it can be concluded that the dependence of ozone in CB6r1 on the isoprene emission is different from those in CB6r2 and CB6r3, due to the change in the products of Reaction (R157), which will be discussed further in a later context.

The $NO_x$ sensitivity to the CB6r3 mechanism is displayed in Fig. 3. It is seen that in CB6r3, except the reactions representing emissions and depositions, chemical reactions (R1) $NO_2 + h\nu \rightarrow NO + O$, (R3) $O_3 + NO \rightarrow NO_2$ and (R26) $NO_2 + O_3 \rightarrow NO_3$, are the most determining reactions for the change of $NO_x$. It is because that the formation of $NO_3$ in the presence of $O_3$ is a major chemical pathway for the loss of $NO_x$, especially in the nighttime. Other important chemical reactions for the change of $NO_x$ include (R9) $O_3 + h\nu \rightarrow O(^1D)$, (R10) $O(^1D) + M \rightarrow O + M$, (R11) $O(^1D) + H_2O \rightarrow 2OH$, and (R25)

$HO_2 + NO \rightarrow OH + NO_2$, which are related to the formation of hydroxyl radicals (i.e. OH). It is not surprising as the reaction between OH and $NO_2$ that forms $HNO_3$ acts as a large sink of reactive nitrogen oxides. Regarding to the other reactions in the mechanism, their sensitivities are much smaller, thus bringing a negligible influence on the change of $NO_x$. With respect to the surface emissions, it can be found in Fig. 3 that the emissions of NO and $NO_2$ would elevate the predicted level of $NO_x$, which is natural. In contrast, the dependence of $NO_x$ on the isoprene emission is minor in this scenario using CB6r3, indicated by

the corresponding small sensitivity (see Reaction (R232) in Fig. 3d). Regarding to the reactions signifying dry depositions, it was found in Fig. 3(d) that the dry deposition of ozone is able to strongly elevate the predicted level of $NO_x$. The reason is that ozone is critical for the conversion of $NO_2$ to $NO_3$, which is a major loss of $NO_x$ in this scenario as discussed above. Thus, the decline of ozone due to dry deposition would substantially inhibit the formation of $NO_3$, thus elevating the concentration of $NO_x$.

The sensitivities of $NO_x$ to the other two CB6 mechanisms, CB6r2 and CB6r1, are shown in Figs. S3 and S4 of the supplementary material, respectively. It was found that the sensitivity of $NO_x$ to CB6r2 shown in Fig. S3 is strongly similar to the sensitivity to CB6r3 displayed in Fig. 3. The largest change in the $NO_x$ sensitivity between CB6r3 and CB6r2 is the addition of reactions representing the temperature dependence of alkyl nitrate formation in CB6r3, i.e. Reactions (R217)-(R220) in Fig. 3(d). However, as mentioned above, the scheme for the temperature dependence in CB6r3 is equivalent to that in CB6r2

under the situation used in this study. Thus, adding these reactions into CB6r3 would not exert a significant influence on the change of $NO_x$. But the moderate sensitivity coefficients belonging to Reactions (R219) and (R220) shown in Fig. 3(d) denote that under a different temperature condition, the change of $NO_x$ brought about by this update might be larger. Apart from this change, other reactions that largely modified between CB6r3 and CB6r2 possess a small sensitivity coefficient. Thus, choosing CB6r3 or CB6r2 in this weak-emission scenario would not significantly influence the predicted $NO_x$. However, for the CB6r1

mechanism, Fig. S4(d) shows that the $NO_x$ sensitivity to the isoprene emission in CB6r1 is substantially larger than that in CB6r2 or CB6r3. It indicates that the dependence of $NO_x$ on the isoprene emission in CB6r1 is heavier than that in CB6r2 and

CB6r3. The reason is also attributed to the change of Reaction (R157), which is similar to the conclusion achieved in the ozone sensitivity analysis discussed above. In CB6r1, the product INTR is generated through Reaction (R157) rather than NTR2 in CB6r2 and CB6r3. As a result, in CB6r1, the emitted isoprene can be more converted to organic compounds such as NTR.

Then, the photolysis of NTR, i.e. Reaction (R92) $NTR + h\nu \rightarrow NO_2 + XO2H + RO2$, would increase the $NO_x$ concentration, leading to a higher $NO_x$ prediction of CB6r1 than those of CB6r2 and CB6r3. This mechanism is also indicated by the increased $NO_x$ sensitivity possessed by Reaction (R92) in CB6r1 compared to that in CB6r2 and CB6r3 (see Fig. S4).

At last, for the HCHO sensitivity, it is seen in Fig. 4 that in CB6r3, the largest HCHO decay pathway is the dry deposition, denoted by its most negative sensitivity coefficient (see Fig. 4d). And another major chemical pathway for the destruction of

HCHO is the photolysis of HCHO, i.e. Reactions (R97) $HCHO + h\nu \rightarrow 2HO_2 + CO$ and (R98) $HCHO + h\nu \rightarrow CO$, both of which possess relatively large absolute values of the HCHO sensitivity. In contrast, major HCHO formation pathways are found including Reactions (R72) $MEO2 + HO_2 \rightarrow 0.9MEPX + 0.1HCHO$ and (R124) $CH_4 + OH \rightarrow MEO2 + RO2$. It is due to the large amount of $CH_4$ in the initial condition, which is a major source of HCHO through its oxidation. Moreover, it was also found in Fig. 4(c) that Reaction (R156) $ISOP + O_3 \rightarrow 0.6HCHO + other\ products$ plays an important role in the

formation of HCHO, due to the emission of isoprene in this scenario. This strongly enhanced HCHO formation by the release of isoprene is also demonstrated by the large positive sensitivity coefficient possessed by the isoprene emission, i.e. (R232) in Fig. 4(d). Thus, it can be concluded that except the initial amount of $CH_4$, the emission intensity of isoprene is also a critical factor determining the predicted value of HCHO in this weak emission scenario.

By comparing the HCHO sensitivity to CB6r3 shown in Fig. 4 with the HCHO sensitivities to CB6r2 (Fig. S5 of the

supplements) and CB6r1 (Fig. S6 of the supplements), it was found that the largest change in the HCHO sensitivity between CB6r3 and CB6r2 is again the addition of reactions representing the temperature dependence in CB6r3. However, similar to the findings discussed above, the prediction of HCHO by CB6r3 is not heavily affected by the addition of these reactions under a standard condition and thus is similar to that by CB6r2. Most interestingly, different from the situations in simulating ozone and $NO_x$, it was found in Fig. 4(d) that the reactions representing the temperature dependence (i.e. Reactions (R217)-(R220))

possess relatively small HCHO sensitivities (<0.05). Thus, it can be expected that even under a different temperature condition, the influence on the HCHO prediction caused by the change of the temperature is also possibly small.

With respect to the HCHO sensitivity to CB6r1 (see Fig. S6 of the supplements), some changes were found. First, different from the small negative sensitivity to Reaction (R157) in CB6r2 and CB6r3, the HCHO sensitivity to Reaction (R157) in CB6r1 has a moderate positive value. We figured out that it is also caused by the difference in the product of this reaction between

CB6r1 and the other two mechanisms. Through Reaction (R157), the emitted isoprene in CB6r1 can be more conveniently converted to active organic compounds such as NTR that can be oxidized and generate HCHO. Therefore, the predicted HCHO in CB6r1 is higher than that in CB6r2 and CB6r3 in this scenario (shown in Fig. 1c). This conclusion is also confirmed by the increased importance of Reaction (R92) $NTR + h\nu \rightarrow NO_2 + XO2H + RO2$ in CB6r1 for the change of HCHO (see Fig. S6), and the elevated HCHO sensitivity to the isoprene emission in CB6r1 ($\sim 0.53$) compared to that in CB6r2 and CB6r3

($\sim 0.48$). Another special finding from the analysis of the HCHO sensitivity to CB6r1 is that in CB6r1, a reverse in the signs of HCHO sensitivities to (R1) $NO_2 + h\nu \rightarrow NO + O$ and (R3) $O_3 + NO \rightarrow NO_2$ occurs (see Fig. S6), compared to those in

CB6r2 and CB6r3. The reason is also attributed to the difference in the HCHO dependence on the isoprene emission between CB6r1 and the other two mechanisms. The occurrence of Reaction (R3) is able to increase the $NO_2$/NO ratio, which further promotes the formation of $NO_3$. Because the reaction between $NO_3$ and isoprene, i.e. Reaction (R157), plays a more important role in the prediction of HCHO in CB6r1 than in CB6r2/CB6r3 as discussed above, the occurrence of Reaction (R3) in CB6r1 can thus remarkably promote the formation of HCHO by accelerating the isoprene+$NO_3$ reaction. In contrast, in CB6r2 or CB6r3, the reaction between isoprene and $NO_3$ plays a relatively minor role in determining HCHO due to the product update from INTR to NTR2. Therefore, instead of accelerating the HCHO formation, $NO_2$ formed through Reaction (R3) in CB6r2 and CB6r3 consumes OH and thus suppresses the formation of HCHO by retarding the oxidation of $CH_4$. As a consequence, the sensitivity of HCHO to Reaction (R3) becomes negative in CB6r2 and CB6r3, shown in Fig. S5 of the supplements and Fig. 4 of the manuscript, respectively.

In general, the sensitivity analysis shows that when the surface emission is weak, One of the updates in CB6r2 and CB6r3 compared with CB6r1 that can strongly affect the ozone prediction is the change in the sink of CXO3, i.e. Reaction (R66). Because of this modification, the significance of many CXO3 related reactions also changes, causing a lower ozone prediction of CB6r2 and CB6r3 than that of CB6r1. Apart from that, the lower ozone predicted by CB6r2 and CB6r3 was also found contributed by the weaker dependence of ozone on the isoprene emission in CB6r2 and CB6r3 than that in CB6r1, due to the change in the product of the reaction between isoprene and $NO_3$. On the contrary, the ozone sensitivities to CB6r2 and CB6r3 are approximately the same, thus leading to a similar $O_3$ prediction. With respect to $NO_x$ and HCHO, it was found that the difference in the product of the reaction between isoprene and $NO_3$ causes a stronger dependence of $NO_x$ and HCHO on the emission of isoprene in CB6r1 than that in CB6r2 or CB6r3. As the isoprene emission in this scenario promotes the formation of $NO_x$ and HCHO, the levels of $NO_x$ and HCHO given by CB6r1 are thus slightly higher than those estimated by CB6r2 and CB6r3. In contrast, reactions that largely modified between CB6r2 and CB6r3 mostly have small sensitivities so that these updates exert a negligible impact on the predictions of $NO_x$ and HCHO, and the predictions given by CB6r2 and CB6r3 are thus similar. However, under a different temperature condition, the predictions of $O_3$ and $NO_x$ by CB6r3 might be largely different from those predicted by CB6r2 and CB6r1, indicated by the moderate values of the sensitivities belonging to the temperature dependent reactions in CB6r3. In contrast, the estimation of HCHO might not be significantly affected, according to the small HCHO sensitivities to these temperature-dependent reactions in CB6r3.

### 3.3  Temporal evolution of ozone, $NO_x$ and HCHO (strong emission)

According to the discussions above, it is known that the difference between the estimations of atmospheric constituents using different CB6 mechanisms is not only caused by the change in the forms of reactions between these mechanisms, but also determined by the different dependence of the mechanism on surface emissions and dry depositions. Thus, we continued to increase the intensity of the surface emission to investigate the behavior of these CB6 mechanisms under a strong emission condition. The emission intensity for each chemical species in this strong emission scenario has been given in Tab. 1.

The temporal profiles of $O_3$, $NO_x$ and HCHO in the strong emission scenario is shown in Fig. 5. It is seen from Fig. 5(a) that after the increase of the surface emission, the ozone concentration keeps steady during the whole simulated period, instead

of dropping to a low level in the weak emission scenario. The ozone level at the end of the 7-day simulation is within a range of 60-130 ppb, much higher than that in the weak emission scenario ($\sim$20-40 ppb). By comparing ozone profiles obtained by using different CB6 mechanisms in Fig. 5(a), we found the ozone predictions by CB6r2 and CB6r3 approximately the same, while CB6r1 predicts a much higher value. It was calculated that the averaged ozone over the 7-th day predicted by CB6r1 is approximately 24 ppb higher than that predicted by CB6r2 or CB6r3. Thus, after the increase of the surface emission, the deviation in the predicted ozone between CB6r1 and the other two CB6 mechanisms is enlarged, compared with that in the weak-emission scenario. It demonstrates that the CB6r1 mechanism has a stronger transformation ability in converting the surface emission into the change of ozone than the other two mechanisms. As a result, simulations using CB6r1 would yield a much higher ozone than that using CB6r2 or CB6r3, even though a same intensity of the surface emission is applied, and the deviation in the predicted ozone would become larger when the applied surface emission increases in the model. In a previous regional modeling of the air quality across the continental US (Ruiz Hildebrandt and Yarwood, 2013), it was reported that CB6r1 predicts a higher ozone than CB6r2. Thus, the results of the present study are consistent with the conclusions of Ruiz Hildebrandt and Yarwood (2013).

The change of $NO_x$ with time is displayed in Fig. 5(b). It shows that $NO_x$ declines rapidly from the relatively high initial value (7 ppb) to a stable level, 1-2 ppb. This final value range is also much higher than that in the weak-emission scenario (<0.5 ppb). An obvious diurnal variation of $NO_x$ is exhibited, and a peak value was found in the early morning of each day. Figure 5(b) also shows that CB6r2 and CB6r3 give similar $NO_x$ predictions, while CB6r1 behaves differently. CB6r1 predicts a higher $NO_x$ than CB6r2 and CB6r3 in most of the simulated period, and the difference grows when the end of the simulation approaches.

With respect to HCHO, we found that due to the inclusion of the strong surface emission in this scenario, the level of HCHO keeps increasing when the simulation proceeds. The emitted species that are responsible for the enhancement of HCHO will be investigated in a later context. The temporal change of HCHO shows a strong diurnal variation, in which it peaks in the afternoon and reaches the trough in the early morning of every day. The predicted HCHO profiles using CB6r2 and CB6r3 are found almost identical. In a box model study of Marvin et al. (2017), they also found that using CB6r3 causes a negligible impact (<1%) on the simulated HCHO compared to using CB6r2, which is consistent with the findings of the present study. In contrast to that, CB6r1 consistently yields a higher value of HCHO than CB6r2 and CB6r3, and the deviation is more pronounced during the daytime. At the 7-th day of the simulation, the peak value of HCHO at noon obtained by CB6r1 is around 14-15 ppb, while the lowest value is approximately 7 ppb. The deviation between the peak values of HCHO predicted by CB6r1 and the other two mechanisms is approximately 2.5 ppb. These values are all much higher than those in the weak-emission scenario (see Fig. 1c), due to the inclusion of the stronger surface emission.

In summary, due to the inclusion of a stronger surface emission, an enhancement of the predicted $O_3$, $NO_x$ and HCHO was found, compared with the weak-emission scenario. Moreover, simulated results of CB6r2 and CB6r3 are almost identical, while CB6r1 consistently gives higher values of $O_3$, $NO_x$ and HCHO than the other two mechanisms. Moreover, the deviations between the estimations by CB6r1 and the other two CB6 mechanisms in this scenario were found enlarged compared to those

in the weak-emission scenario, reflecting a stronger transformation ability of CB6r1 converting the surface emissions into the change of atmospheric constituents.

## 3.4    Concentration sensitivity analysis of different CB6 mechanisms (strong emission)

The concentration sensitivity analysis was applied on these CB6 mechanisms again, after implementing the strong surface emission. Figure 6 shows the ozone sensitivity to the CB6r3 mechanism. From a global view, it can be found that after increas-

ing the surface emission, the importance of many reactions in the mechanism increases, compared with the weak-emission case. Reactions that the significance changes the most are (1) $NO_x$ related reactions: (R1) $NO_2 + h\nu \rightarrow NO + O$ and (R3) $O_3 + NO \rightarrow NO_2$; (2) terminal olefins (OLE) related reactions: (R142) $OLE + OH \rightarrow 0.78HCHO + 0.49ALD2 + 0.49ALDX + 0.98XO2H + 0.2XO2 + 0.02XO2N + 1.2RO2 - 0.73PAR$ and (R143) $OLE + O_3 \rightarrow 0.29ALD2 + 0.56HCHO + 0.27ALDX + 0.15XO2H + 0.15RO2 + 0.33OH + 0.08HO_2 + 0.38CO + 0.07GLY + 0.07MGLY + 0.09FACD + 0.13AACD + 0.04H_2O_2 - 0.79PAR$;

(3) isoprene related reactions: (R149) $ISOP + OH \rightarrow ISO2 + RO2$ and (R157) $ISOP + NO_3 \rightarrow 0.35NO_2 + 0.65NTR2 + 0.64XO2H + 0.33XO2 + 0.03XO2N + RO2 + 0.35HCHO + 0.35ISPD$. Moreover, Fig. 6(d) shows that the surface emissions of many chemical species exert a strong influence on the change of ozone, reflected by the relatively large ozone sensitivities to Reactions (R222)-(R234). It can be seen in Fig. 6 that the ozone sensitivities to the surface emissions are comparable to those belonging to the $NO_x$ related reactions (i.e. (R1) and (R3)). In addition, similar to the weak-emission scenario, dry deposition,

denoted by Reaction (R233) in Fig. 6(d), acts as a major loss pathway of ozone, indicated by the large absolute value of its ozone sensitivity.

      Among the surface emissions, the most influential emitted species for the change of $O_3$ is NO (see Fig. 6d). It is not surprising as the oxidation of the emitted NO by the hydroperoxy radical and methylperoxy radicals would form $O_3$. Aside from this, the release of $NO_2$ would also increase the $O_3$ level, through its photolytic decomposition. In contrast to the $NO_x$ emissions, the

increase of VOC emissions in this scenario would decrease the formation of $O_3$, which is indicated in Fig. 6(d) by the negative ozone sensitivities to the VOC emissions. Moreover, it was found that the ozone sensitivity to the emissions of $NO_x$ is larger than that to the VOC emissions. This trend has also been revealed by Luecken et al. (2018), showing that the dependence of ozone on $NO_x$ is approximately three times as heavy as that on hydrocarbons in their model studies. In the present study, the most influential VOC for the change of $O_3$ is found as ISOP (isoprene). Isoprene can react rapidly with OH and $NO_3$, which

substantially contributes to the formation and consumption of ozone. Thus, among the emissions of VOCs, more attention should be paid to the isoprene emission applied in air quality models in order to achieve a more accurate ozone prediction. In previous studies, it was shown that different biogenic emission models (e.g. MEGAN and BEIS) may yield different isoprene emission estimates (Bash et al., 2016; Zhang et al., 2017b). Thus, the choice of the biogenic emission model in the settings of the air quality model would strongly influence the modeled ozone, according to the findings of the present study. Moreover, an

enhancement of the emissions of terminal olefins (OLE), toluene (TOL), xylene and other polyalkyl aromatics (XYL) would also reduce the ozone level. From the response of the ozone concentration to the surface emission, we concluded that in this scenario, an emission control of $NO_x$ especially NO is effective in reducing $O_3$, while an emission control of VOCs leads to an increase of ozone, when CB6r3 is implemented.

The ozone sensitivities to the reactions of CB6r2 and CB6r1 as well as the surface emissions and dry depositions are shown in Figs. S7 and S8 of the supplementary material. First, by comparing Fig. 6 with Fig. S7 in the supplements, we found that after the inclusion of the strong surface emission, the ozone sensitivities to CB6r3 and CB6r2 are still approximately the same. As a result, under this condition, $O_3$ predicted by CB6r3 is almost equal to that predicted by CB6r2, which has been displayed in Fig 5(a). For the CB6r1 mechanism, the ozone sensitivity displayed in Fig. S8 of the supplements shows a remarkable difference, especially in the dependence of $O_3$ on the surface emission. It was found that in this strong-emission scenario, the dependence of $O_3$ on the emission of isoprene is weaker in CB6r1 than that in CB6r2 and CB6r3 (see the sensitivity to Reaction (R233) in Fig. S8 of the supplements). On the contrary, the ozone sensitivities to the emissions of other hydrocarbons in CB6r1 mostly keep unchanged. Thus, in CB6r1, the emitted VOC that $O_3$ depends on the most becomes XYL (xylene and other polyalkyl aromatics) instead of ISOP (isoprene). As a result, the $O_3$ destruction caused by the isoprene emission is suppressed in CB6r1. The importance of many isoprene related reactions, such as Reaction (R149) $ISOP+OH \rightarrow ISO2+RO2$ also becomes weaker in CB6r1 than that in CB6r2 and CB6r3. The reason for the less dependence of ozone on the isoprene emission in CB6r1 under this strong emission condition is again attributed to the change in the product of Reaction (R157), i.e. $ISOP + NO_3$. As discussed above in the weak-emission scenario, Reaction (R157) in CB6r1 is able to promote the formation of ozone by converting the emitted isoprene into organic nitrates such as INTR and NTR, which can be more conveniently recycled than the product NTR2 in CB6r2 and CB6r3. Because in this strong-emission scenario, the release of VOCs tends to suppress the ozone formation, the positive influence of the isoprene emission on the change of ozone brought by Reaction (R157) in CB6r1 can thus offset the negative dependence of ozone on the emission of isoprene, leading to a weaker dependence of ozone in CB6r1. This positive influence on the formation of ozone caused by Reaction (R157) in CB6r1 is also reflected by the shift of the ozone sensitivity to Reaction (R157) from negative in CB6r2 and CB6r3 to posivite in CB6r1 (see Fig. S8), as well as the relatively large sensitivities possessed by INTR and NTR related reactions, i.e. (R170) $INTR + OH$ and (R92) $NTR + h\nu$ in CB6r1. Therefore, under the condition of a strong surface emission, due to the less dependence of $O_3$ on the emitted isoprene, the $O_3$ level predicted by CB6r1 is higher than that predicted by CB6r2 and CB6r3, which has been shown in Fig 5(a). In the weak-emission scenario shown before, we found the modification in Reaction (R66) about the sink of $CXO_3$ to be another major factor causing the difference between the simulation results of CB6r1 and CB6r2/CB6r3. However, it can be seen in Fig. S8 that after increasing the surface emission, the ozone sensitivity to Reaction (R66) is negligible, compared to the sensitivity to the surface emission. Therefore, the difference in the predicted ozone between different CB6 mechanisms is mostly caused by the change of the $O_3$ dependence on the surface emission, especially the release of isoprene.

The sensitivities of $NO_x$ to the reactions of the CB6r3 mechanism as well as the surface emissions and dry depositions are displayed in Fig. 7. It can be seen that after the increase of the surface emission, the most dominant factor for the change of $NO_x$ is still the direct emission of $NO_x$ (i.e. Reactions (R222) and (R223) in Fig. 7d). Moreover, due to the enhanced $NO_x$ emission in this scenario, the significance of Reactions (R1) $NO_2 + h\nu \rightarrow NO + O$, (R3) $O_3 + NO \rightarrow NO_2$ and (R26) $NO_2 + O_3 \rightarrow NO_3$, which represent the inter-conversion of reactive nitrogen oxides and the loss of $NO_x$ due to the formation of $NO_3$, increases, compared with that in the weak-emission scenario. In contrast, for the reactions associated with the formation of OH: (R9) $O_3 + h\nu \rightarrow O(^1D)$, (R10) $O(^1D) + M \rightarrow O + M$, (R11) $O(^1D) + H_2O \rightarrow 2OH$, and (R25) $HO_2 + NO \rightarrow OH + NO_2$, which

used to be important in the weak-emission scenario, their $NO_x$ sensitivities decrease. It is because that after the enhancement of the surface emission, the ozone level is elevated, so that $NO_2$ is more involved in Reaction (R26) $NO_2 + O_3 \rightarrow NO_3$. As a result, the importance of the reaction between $NO_2$ and OH for the change of $NO_x$ drops, leading to a less dependence of $NO_x$ on the formation of OH under this situation.

In this strong-emission scenario using CB6r3, the emitted species that $NO_x$ depends on the most is NO (see Fig. 7d), which is natural as the direct emission of NO would strongly increase $NO_x$. The release of $NO_2$ also elevates the $NO_x$ level predicted by CB6r3. In contrast to that, the release of different VOCs exerts a different influence on the concentration change of $NO_x$. The emissions of ISO (isoprene) and OLE (terminal olefins) would increase the simulated level of $NO_x$, while the emissions of TOL (toluene) and XYL (xylene and other polyalkyl aromatics) decrease it. Figure. 7(d) also shows that the dependence of $NO_x$ on the emissions of VOCs is remarkably lower than that on the direct emissions of $NO_x$, which is similar to the conclusion achieved in the ozone sensitivity analysis. In addition, Fig. 7(d) shows that in this scenario, the dry deposition of ozone still plays the role of enhancing the formation of $NO_x$ as that in the weak-emission scenario, due to the deceleration of $NO_3$ formation by the decline of ozone.

From a comparison between the $NO_x$ sensitivities to CB6r2 (shown in Fig. S9 of the supplements) and CB6r3 (see Fig. 7), again we found that the $NO_x$ sensitivity to CB6r2 is almost identical to that to CB6r3, thus leading to a similar prediction of $NO_x$ by these two mechanisms. However, for CB6r1 (see Fig. S10 of the supplements), the sensitivities of $NO_x$ to the surface emissions of NO and $NO_2$ (0.84 and 0.13) are relatively larger than those in CB6r2/CB6r3 (0.76-0.79 and 0.11). The reason for the heavier dependence of $NO_x$ on the surface emissions in CB6r1 is that the reaction between isoprene and $NO_3$ in CB6r1 facilitates the formation of organic nitrates (i.e. NTR, INTR) as discussed above. The formed organic nitrates are then photolyzed or react with OH, forming $NO_x$. As a result, the emitted $NO_x$ is more involved in the chemical system represented by CB6r1, leading to a higher dependence of $NO_x$ on the direct release of NO and $NO_2$ in CB6r1. As a consequence, the $NO_x$ level predicted by CB6r1 is higher than those given by CB6r2 and CB6r3 with the same emission intensity of $NO_x$.

At last, we focused on the averaged sensitivities of HCHO to these three different CB6 mechanisms. Figure 8 shows the HCHO sensitivities to the chemical reactions, surface emissions and dry depositions for the CB6r3 mechanism. It was found that processes that play an important role in the change of HCHO include (R96) $HCHO + OH \rightarrow HO_2 + CO$, (R97) $HCHO + h\nu \rightarrow 2HO_2 + CO$, and (R98) $HCHO + h\nu \rightarrow CO$, which are reactions consuming HCHO, and the dry deposition process denoted by Reaction (R243) in Fig. 8(d). These important HCHO decay pathways are the same to those in the weak-emission case. In addition, a strong negative role of Reaction (R26), $NO_2 + O_3 \rightarrow NO_3$, in determining HCHO is also identified under this strong emission condition (see Fig. 8a). The reason is that in CB6r3, the emitted isoprene reacts with $NO_3$ generated from Reaction (R26), forming NTR2. NTR2 is then converted to the inactive $HNO_3$ and thus exerts a minor influence on the change of HCHO. Therefore, the occurrence of Reaction (R26) in CB6r3 can substantially suppress the formation of HCHO caused by the release of isoprene, especially under this strong $NO_x$ emission condition. It is also interesting to note that Reaction (R124) $CH_4 + OH \rightarrow MEO2 + RO2$ that used to strongly promote the formation of HCHO in the weak-emission scenario no longer possesses a large sensitivity. It denotes a decreased importance of the initial $CH_4$ in the formation of HCHO after increasing the surface emission in the model. Instead, the release of VOCs would significantly promote the formation of HCHO.

From the dependence of HCHO on the surface emissions displayed in Fig. 8(d), we found that an increase in the emission
intensity of VOCs especially isoprene and terminal olefins would significantly enhance the HCHO formation, and the influence
caused by the emissions of isoprene and terminal olefins is approximately equal, indicated by the similar sensitivities to these
two emissions. This strong influence of the emissions of isoprene and other olefins on the change of HCHO has also been
identified in many previous studies (Luecken et al., 2006, 2008; Wolfe et al., 2016; Marvin et al., 2017). Moreover, in the
present study, we found that the increase of the $NO_x$ emission leads to an elevation of HCHO. It is because that the release of
$NO_x$ would significantly increase the ozone level in this scenario. As Reaction (R156) $ISOP + O_3 \rightarrow 0.6HCHO + 0.65ISPD +$
$0.15ALDX + 0.2CXO3 + 0.35PAR + 0.27OH + 0.2XO2 + 0.2RO2 + 0.07HO_2 + 0.07CO$ is a major pathway for the formation of
HCHO, reflected by the strong HCHO dependence on Reaction (R156) (see Fig. 8c), the increase of ozone due to the enhanced
$NO_x$ emission would thus promote the formation of HCHO, leading to a positive dependence of HCHO on the release of $NO_x$.
It was also found in this study that HCHO is more sensitive to VOCs than $NO_x$, which is in accordance with the conclusions
achieved in the previous sensitivity study of HCHO to precursor species (Luecken et al., 2018). The reason is that HCHO can
be formed under both $NO_x$-rich and $NO_x$-poor conditions, resulting in a weaker dependence of HCHO on the $NO_x$ emissions
than that on the VOC emissions.

In a previous modeling study conducted by Luecken et al. (2019) using CB6r3, they found an underestimation of HCHO in
a comparison with observations across the US. Luecken et al. (2019) suggested that the underestimation of HCHO might be
caused by the uncertainties in biogenic emissions including direct HCHO emissions and other VOC emissions. Based on our
findings, we suggested that the underestimation of HCHO might be caused by the underestimation of isoprene and other alkene
emissions. In contrast, the direct emission of HCHO may possibly only exert a minor impact on the change of HCHO, according
to the sensitivity analysis of CB6r3 in the present study. In the study of Luecken et al. (2019), they also performed a sensitivity
test by doubling the isoprene emission, and it was found that the simulated HCHO is elevated due to the enhancement of the
isoprene emission. This is also in accordance with our findings in the present study.

By comparing the HCHO sensitivities to CB6r3 (Fig 8) with those to CB6r2 (Fig. S11 in the supplements), we noticed that
the most dominant reactions for the change of HCHO approximately the same in these two mechanisms, thus leading to a
similar HCHO prediction by these two mechanisms. However, the sensitivity of HCHO to CB6r1 displayed in Fig. S12 of the
supplements shows that in CB6r1, the significance of the isoprene emission on the change of HCHO ($\sim$0.39) is approximately
10% higher than that in CB6r2 and CB6r3 ($\sim$0.36). As a result, a same increment in the isoprene emission would lead to a
relatively larger increase in HCHO predicted by CB6r1 relative to that predicted by CB6r2 or CB6r3. This is also the reason
for the relatively higher HCHO prediction by CB6r1 shown in Fig. 5. Again, it is attributed to the higher ozone prediction of
CB6r1 caused by the change of Reaction (R157). Because the major HCHO formation pathway in this simulation scenario
is the reaction between isoprene and ozone, under the condition with higher ozone estimated by CB6r1, larger amount of the
emitted isoprene can be converted to HCHO, thus causing the higher HCHO prediction and the heavier dependence of HCHO
on the isoprene emission in CB6r1. This finding again indicates the importance of revising the isoprene chemistry in future
updates to the CB6 mechanism.

In summary, in the situation with the inclusion of the strong surface emission in the model, we found that the ozone level predicted by CB6r2 and CB6r3 depends heavily on the surface emission especially the release of NO and isoprene. In contrast, the dependence of ozone on the isoprene is weaker in CB6r1. Aside from that, the importance of many isoprene related reactions in CB6r1 for the change of ozone decreases, which is shown in the sensitivity analysis. These changes in the ozone sensitivity lead to a higher ozone prediction of CB6r1 relative to that of CB6r2 and CB6r3, even though a same surface emission condition is applied. With respect to the change of $NO_x$, in CB6r2 and CB6r3, the most influential emissions are also the release of NO and isoprene. However, in CB6r1, the dependence of the predicted $NO_x$ on the surface emissions of NO and $NO_2$ is heavier than that in CB6r2 and CB6r3, due to the change in the product of the reaction between isoprene and $NO_3$, resulting in a higher $NO_x$ prediction of CB6r1. At last, for HCHO, the sensitivity analysis shows that the change of HCHO relies more on the emissions of VOCs instead of the emissions of $NO_x$, and the enhancement of the VOC emissions particularly isoprene and terminal olefins would significantly promote the formation of HCHO. However, the dependence of HCHO on the release of isoprene in CB6r1 is stronger than that in the other two mechanisms, thus leading to a higher prediction of HCHO in CB6r1 under the same emission condition.

## 4   Conclusions and Future Work

In the present study, we found that different versions of the CB6 mechanism perform differently in simulating $O_3$, $NO_x$ and HCHO, although the same initial condition and the same intensity of the surface emission is set up. When the surface emission is weak, CB6r1 predicts a higher ozone value than the other two mechanisms, and the deviation is approximately 7 ppb. The sensitivity analysis suggests that the higher ozone prediction by CB6r1 is partly caused by the modification of the chemical loss pathways of acylperoxy radicals with three and higher carbons (i.e. species CXO3) in the mechanism. Due to this modification, less CXO3 is consumed in CB6r1 than that in CB6r2 and CB6r3, resulting in a higher ozone prediction by CB6r1. Moreover, the ozone sensitivity to the isoprene emission in CB6r1 was found larger than that in CB6r2 and CB6r3, which also contributes to the higher ozone prediction of CB6r1 under the same isoprene emission condition. Regarding to $NO_x$ and HCHO, the estimations given by CB6r1 are higher than those given by CB6r2 and CB6r3, but the deviations between the simulation results become smaller during the end of the 7-day computation. The sensitivity analysis also shows that the update in CB6r3 about the temperature dependence of organic nitrate formation might exert a strong influence on the predictions of ozone and $NO_x$ under a different temperature condition, while the impact of the temperature change on HCHO might be minor.

After implementing a strong surface emission into the model, we found the simulated levels of $O_3$, $NO_x$ and HCHO elevated, compared with those in the weak-emission scenario. It was also found that the ozone concentration predicted by CB6r2 and CB6r3 depends on the emissions of NO and isoprene the most, while in CB6r1 the dependence of ozone on the isoprene emission is weaker. Because in this simulation scenario, the isoprene emission tends to suppress the ozone formation, ozone predicted by CB6r1 is thus higher than those predicted by CB6r2 and CB6r3 with the same emission intensity. With respect to the $NO_x$ prediction, in CB6r1, the association between the mixing ratio of $NO_x$ and the release of NO and $NO_2$ was found stronger than that in CB6r2 and CB6r3. It is because that the released $NO_x$ is more involved in the reaction system represented

by CB6r1, thus leading to a higher NO$_x$ prediction of CB6r1, compared with that given by CB6r2 or CB6r3. At last, we found that the HCHO predictions of these three CB6 mechanisms rely mostly on the emissions of NO, isoprene and terminal olefins. However, in CB6r1, the association between HCHO and the isoprene emission is stronger, resulting in a higher HCHO prediction relative to that in CB6r2 and CB6r3 with the same isoprene emission.

The present study has its limitations. The conclusions achieved in this study are mostly valid under conditions that have been presented and analyzed in this box model study, and these conditions may possibly differ from those present in 3-D model simulations of the atmosphere. Therefore, in the future, we plan to test the behavior of these CB6 mechanisms under different environmental conditions with different surface emission intensities, epscially the conditions that are implemented in 3-D model studies of the atmosphere. The influence caused by the varying of the temperature on the concentration change

of the focused species, especially for CB6r3 should also be investigated. Moreover, the latest version of the CB6 mechanism, CB6r4 (Emery et al., 2016), should be studied and compared with the three CB6 mechanisms investigated in the present study, particularly in a halogen-rich environment. In addition, the conclusions achieved in this box-model study need to be confirmed in simulations using multi-dimensional air quality models. At present, we are conducting three-dimensional simulations using CMAQ (Byun and Schere, 2006) and CAMx (ENVIRON, 2015; Ramboll Environment and Health, 2020) to discover the

difference in modeling O$_3$, NO$_x$ and HCHO by using these different versions of the CB6 mechanism, which is attributed to a future publication.

*Code and data availability.*  The source code of the model and the data of the computational results shown in this article can be acquired from the link: https://pan.baidu.com/s/1Gi_Tb-SIrMi0IvD-4FBYJQ, using the password: bpc1.

**Appendix A**

**Table A1.** Complete listings of chemical reactions belonging to different CB6 mechanisms used in the present study. The updates between different versions of the CB6 mechanism are also marked. The abbreviation "-" denotes that there is no change in the form of this reaction between different CB6 mechanisms.

| Reaction Number | CB6r1[a] | Reaction Number | CB6r2[b] | Reaction Number | CB6r3[c] |
|---|---|---|---|---|---|
| (R1) | $NO_2 + h\nu \rightarrow NO + O$ | (R1) | - | (R1) | - |
| (R2) | $O + O_2 + M \rightarrow O_3 + M$ | (R2) | - | (R2) | - |
| (R3) | $O_3 + NO \rightarrow NO_2$ | (R3) | - | (R3) | - |
| (R4) | $O + NO + M \rightarrow NO_2 + M$ | (R4) | - | (R4) | - |
| (R5) | $O + NO_2 \rightarrow NO$ | (R5) | - | (R5) | - |
| (R6) | $O + NO_2 \rightarrow NO_3$ | (R6) | - | (R6) | - |
| (R7) | $O + O_3 \rightarrow$ | (R7) | - | (R7) | - |
| (R8) | $O_3 + h\nu \rightarrow O$ | (R8) | - | (R8) | - |
| (R9) | $O_3 + h\nu \rightarrow O(^1D)$ | (R9) | - | (R9) | - |
| (R10) | $O(^1D) + M \rightarrow O + M$ | (R10) | - | (R10) | - |

| Reaction Number | CB6r1[a] | Reaction Number | CB6r2[b] | Reaction Number | CB6r3[c] |
|---|---|---|---|---|---|
| (R11) | $O(^1D) + H_2O \rightarrow 2OH$ | (R11) | - | (R11) | - |
| (R12) | $O_3 + OH \rightarrow HO_2$ | (R12) | - | (R12) | - |
| (R13) | $O_3 + HO_2 \rightarrow OH$ | (R13) | - | (R13) | - |
| (R14) | $OH + O \rightarrow HO_2$ | (R14) | - | (R14) | - |
| (R15) | $HO_2 + O \rightarrow OH$ | (R15) | - | (R15) | - |
| (R16) | $OH + OH \rightarrow O$ | (R16) | - | (R16) | - |
| (R17) | $OH + OH \rightarrow H_2O_2$ | (R17) | - | (R17) | - |
| (R18) | $OH + HO_2 \rightarrow$ | (R18) | - | (R18) | - |
| (R19) | $HO_2 + HO_2 \rightarrow H_2O_2$ | (R19) | - | (R19) | - |
| (R20) | $HO_2 + HO_2 + H_2O \rightarrow H_2O_2$ | (R20) | - | (R20) | - |
| (R21) | $H_2O_2 + h\nu \rightarrow 2OH$ | (R21) | - | (R21) | - |
| (R22) | $H_2O_2 + OH \rightarrow HO_2$ | (R22) | - | (R22) | - |
| (R23) | $H_2O_2 + O \rightarrow OH + HO_2$ | (R23) | - | (R23) | - |
| (R24) | $NO + NO + O_2 \rightarrow 2NO_2$ | (R24) | - | (R24) | - |
| (R25) | $HO_2 + NO \rightarrow OH + NO_2$ | (R25) | - | (R25) | - |
| (R26) | $NO_2 + O_3 \rightarrow NO_3$ | (R26) | - | (R26) | - |
| (R27) | $NO_3 + h\nu \rightarrow NO_2 + O$ | (R27) | - | (R27) | - |
| (R28) | $NO_3 + h\nu \rightarrow NO$ | (R28) | - | (R28) | - |
| (R29) | $NO_3 + NO \rightarrow 2NO_2$ | (R29) | - | (R29) | - |
| (R30) | $NO_3 + NO_2 \rightarrow NO + NO_2$ | (R30) | - | (R30) | - |
| (R31) | $NO_3 + O \rightarrow NO_2$ | (R31) | - | (R31) | - |
| (R32) | $NO_3 + OH \rightarrow HO_2 + NO_2$ | (R32) | - | (R32) | - |
| (R33) | $NO_3 + HO_2 \rightarrow OH + NO_2$ | (R33) | - | (R33) | - |
| (R34) | $NO_3 + O_3 \rightarrow NO_2$ | (R34) | - | (R34) | - |
| (R35) | $NO_3 + NO_3 \rightarrow 2NO_2$ | (R35) | - | (R35) | - |
| (R36) | $NO_3 + NO_2 \rightarrow N_2O_5$ | (R36) | - | (R36) | - |
| (R37) | $N_2O_5 \rightarrow NO_3 + NO_2$ | (R37) | - | (R37) | - |
| (R38) | $N_2O_5 + h\nu \rightarrow NO_3 + NO_2$ | (R38) | - | (R38) | - |
| (R39) | $N_2O_5 + H_2O \rightarrow 2HNO_3$ | (R39) | - | (R39) | - |
| (R40) | $NO + OH \rightarrow HONO$ | (R40) | - | (R40) | - |

| Reaction Number | CB6r1[a] | Reaction Number | CB6r2[b] | Reaction Number | CB6r3[c] |
|---|---|---|---|---|---|
| (R41) | $NO + NO_2 + H_2O \rightarrow 2HONO$ | (R41) | - | (R41) | - |
| (R42) | $HONO + HONO \rightarrow NO + NO_2$ | (R42) | - | (R42) | - |
| (R43) | $HONO + h\nu \rightarrow NO + OH$ | (R43) | - | (R43) | - |
| (R44) | $HONO + OH \rightarrow NO_2$ | (R44) | - | (R44) | - |
| (R45) | $NO_2 + OH \rightarrow HNO_3$ | (R45) | - | (R45) | - |
| (R46) | $HNO_3 + OH \rightarrow NO_3$ | (R46) | - | (R46) | - |
| (R47) | $HNO_3 + h\nu \rightarrow OH + NO_2$ | (R47) | - | (R47) | - |
| (R48) | $HO_2 + NO_2 \rightarrow PNA$ | (R48) | - | (R48) | - |
| (R49) | $PNA \rightarrow HO_2 + NO_2$ | (R49) | - | (R49) | - |
| (R50) | $PNA + h\nu \rightarrow 0.59HO_2 + 0.59NO_2 +$ $0.41OH + 0.41NO_3$ | (R50) | - | (R50) | - |
| (R51) | $PNA + OH \rightarrow NO_2$ | (R51) | - | (R51) | - |
| (R52) | $SO2 + OH \rightarrow SULF + HO_2$ | (R52) | - | (R52) | - |
| (R53) | $C2O3 + NO \rightarrow NO_2 + MEO2 + RO2$ | (R53) | - | (R53) | - |
| (R54) | $C2O3 + NO_2 \rightarrow PAN$ | (R54) | - | (R54) | - |
| (R55) | $PAN \rightarrow C2O3 + NO_2$ | (R55) | - | (R55) | - |
| (R56) | $PAN + h\nu \rightarrow 0.6NO_2 + 0.6C2O3 +$ $0.4NO_3 + 0.4MEO2 + 0.4RO2$ | (R56) | - | (R56) | - |
| (R57) | $C2O3 + HO_2 \rightarrow 0.41PACD + 0.15AACD +$ $0.15O_3 + 0.44MEO2 + 0.44RO2 + 0.44OH$ | (R57) | - | (R57) | - |
| (R58) | $C2O3 + RO2 \rightarrow C2O3$ | (R58) | - | (R58) | - |
| (R59) | $C2O3 + C2O3 \rightarrow 2MEO2 + 2RO2$ | (R59) | - | (R59) | - |
| (R60) | $C2O3 + CXO3 \rightarrow MEO2 + ALD2 +$ $XO2H + 2RO2$ | (R60) | - | (R60) | - |
| (R61) | $CXO3 + NO \rightarrow NO_2 + ALD2 +$ $XO2H + RO2$ | (R61) | - | (R61) | - |
| (R62) | $CXO3 + NO_2 \rightarrow PANX$ | (R62) | - | (R62) | - |
| (R63) | $PANX \rightarrow NO_2 + CXO3$ | (R63) | - | (R63) | - |
| (R64) | $PANX + h\nu \rightarrow 0.6NO_2 + 0.6CXO3 +$ $0.4NO_3 + 0.4ALD2 + 0.4XO2H + 0.4RO2$ | (R64) | - | (R64) | - |

| Reaction Number | CB6r1[a] | Reaction Number | CB6r2[b] | Reaction Number | CB6r3[c] |
|---|---|---|---|---|---|
| (R65) | $CXO3 + HO_2 \rightarrow 0.41PACD + 0.15AACD + 0.15O_3 + 0.44ALD2 + 0.44XO2H + 0.44RO2 + 0.44OH$ | (R65) | - | (R65) | - |
| (R66) | $CXO3 + RO2 \rightarrow CXO3$ | (R66) | $CXO3 + RO2 \rightarrow 0.8ALD2 + 0.8XO2H + 0.8RO2$ | (R66) | - |
| (R67) | $CXO3 + CXO3 \rightarrow 2ALD2 + 2XO2H + 2RO2$ | (R67) | - | (R67) | - |
| (R68) | $RO2 + NO \rightarrow NO$ | (R68) | - | (R68) | - |
| (R69) | $RO2 + HO_2 \rightarrow HO_2$ | (R69) | - | (R69) | - |
| (R70) | $RO2 + RO2 \rightarrow$ | (R70) | - | (R70) | - |
| (R71) | $MEO2 + NO \rightarrow HCHO + HO_2 + NO_2$ | (R71) | - | (R71) | - |
| (R72) | $MEO2 + HO_2 \rightarrow 0.9MEPX + 0.1HCHO$ | (R72) | - | (R72) | - |
| (R73) | $MEO2 + C2O3 \rightarrow HCHO + 0.9HO_2 + 0.9MEO2 + 0.1AACD + 0.9RO2$ | (R73) | - | (R73) | - |
| (R74) | $MEO2 + RO2 \rightarrow 0.685HCHO + 0.315MEOH + 0.37HO_2 + RO2$ | (R74) | - | (R74) | - |
| (R75) | $XO2H + NO \rightarrow NO_2 + HO_2$ | (R75) | - | (R75) | - |
| (R76) | $XO2H + HO_2 \rightarrow ROOH$ | (R76) | - | (R76) | - |
| (R77) | $XO2H + C2O3 \rightarrow 0.8HO_2 + 0.8MEO2 + 0.2AACD + 0.8RO2$ | (R77) | - | (R77) | - |
| (R78) | $XO2H + RO2 \rightarrow 0.6HO_2 + RO2$ | (R78) | - | (R78) | - |
| (R79) | $XO2 + NO \rightarrow NO_2$ | (R79) | - | (R79) | - |
| (R80) | $XO2 + HO_2 \rightarrow ROOH$ | (R80) | - | (R80) | - |
| (R81) | $XO2 + C2O3 \rightarrow 0.8MEO2 + 0.2AACD + 0.8RO2$ | (R81) | - | (R81) | - |
| (R82) | $XO2 + RO2 \rightarrow 0.6HO_2 + RO2$ | (R82) | $XO2 + RO2 \rightarrow RO2$ | (R82) | - |
| (R83) | $XO2N + NO \rightarrow NTR$ | (R83) | $XO2N + NO \rightarrow 0.5NTR1 + 0.5NTR2$ | (R83) | - |
| (R84) | $XO2N + HO_2 \rightarrow ROOH$ | (R84) | - | (R84) | - |
| (R85) | $XO2N + C2O3 \rightarrow 0.8HO_2 + 0.8MEO2 + 0.2AACD + 0.8RO2$ | (R85) | - | (R85) | - |
| (R86) | $XO2N + RO2 \rightarrow 0.6HO_2 + RO2$ | (R86) | $XO2N + RO2 \rightarrow RO2$ | (R86) | - |
| (R87) | $MEPX + OH \rightarrow 0.6MEO2 + 0.6RO2 + 0.4HCHO + 0.4OH$ | (R87) | - | (R87) | - |
| (R88) | $MEPX + h\nu \rightarrow MEO2 + RO2 + OH$ | (R88) | - | (R88) | - |
| (R89) | $ROOH + OH \rightarrow 0.54XO2H + 0.06XO2N + 0.6RO2 + 0.4OH$ | (R89) | - | (R89) | - |
| (R90) | $ROOH + h\nu \rightarrow HO_2 + OH$ | (R90) | - | (R90) | - |

| Reaction Number | CB6r1[a] | Reaction Number | CB6r2[b] | Reaction Number | CB6r3[c] |
|---|---|---|---|---|---|
| (R91) | $NTR + OH \rightarrow HNO_3 + XO2H + RO2$ | (R91) | $NTR1 + OH \rightarrow NTR2$ | (R91) | - |
| (R92) | $NTR + h\nu \rightarrow NO_2 + XO2H + RO2$ | (R92) | $NTR1 + h\nu \rightarrow NO_2$ | (R92) | - |
| (R93) | $FACD + OH \rightarrow HO_2$ | (R93) | - | (R93) | - |
| (R94) | $AACD + OH \rightarrow MEO2 + RO2$ | (R94) | - | (R94) | - |
| (R95) | $PACD + OH \rightarrow C2O3$ | (R95) | - | (R95) | - |
| (R96) | $HCHO + OH \rightarrow HO_2 + CO$ | (R96) | - | (R96) | - |
| (R97) | $HCHO + h\nu \rightarrow 2HO_2 + CO$ | (R97) | - | (R97) | - |
| (R98) | $HCHO + h\nu \rightarrow CO$ | (R98) | - | (R98) | - |
| (R99) | $HCHO + O \rightarrow OH + HO_2 + CO$ | (R99) | - | (R99) | - |
| (R100) | $HCHO + NO_3 \rightarrow HNO_3 + HO_2 + CO$ | (R100) | - | (R100) | - |
| (R101) | $HCHO + HO_2 \rightarrow HCO3$ | (R101) | - | (R101) | - |
| (R102) | $HCO3 \rightarrow HCHO + HO_2$ | (R102) | - | (R102) | - |
| (R103) | $HCO3 + NO \rightarrow FACD + NO_2 + HO_2$ | (R103) | - | (R103) | - |
| (R104) | $HCO3 + HO_2 \rightarrow 0.5MEPX + 0.5FACD + 0.2OH + 0.2HO_2$ | (R104) | - | (R104) | - |
| (R105) | $ALD2 + O \rightarrow C2O3 + OH$ | (R105) | - | (R105) | - |
| (R106) | $ALD2 + OH \rightarrow C2O3$ | (R106) | - | (R106) | - |
| (R107) | $ALD2 + NO_3 \rightarrow C2O3 + HNO_3$ | (R107) | - | (R107) | - |
| (R108) | $ALD2 + h\nu \rightarrow MEO2 + RO2 + CO + HO_2$ | (R108) | - | (R108) | - |
| (R109) | $ALDX + O \rightarrow CXO3 + OH$ | (R109) | - | (R109) | - |
| (R110) | $ALDX + OH \rightarrow CXO3$ | (R110) | - | (R110) | - |
| (R111) | $ALDX + NO_3 \rightarrow CXO3 + HNO_3$ | (R111) | - | (R111) | - |
| (R112) | $ALDX + h\nu \rightarrow MEO2 + RO2 + CO + HO_2$ | (R112) | $ALDX + h\nu \rightarrow ALD2 + XO2H + RO2 + CO + HO_2$ | (R112) | - |
| (R113) | $GLYD + OH \rightarrow 0.2GLY + 0.2HO_2 + 0.8C2O3$ | (R113) | - | (R113) | - |
| (R114) | $GLYD + h\nu \rightarrow 0.74HCHO + 0.89CO + 1.4HO_2 + 0.15MEOH + 0.19OH + 0.11GLY + 0.11XO2H + 0.11RO2$ | (R114) | - | (R114) | - |
| (R115) | $GLYD + NO_3 \rightarrow HNO_3 + C2O3$ | (R115) | - | (R115) | - |
| (R116) | $GLY + OH \rightarrow 1.7CO + 0.3XO2 + 0.3RO2 + HO_2$ | (R116) | $GLY + OH \rightarrow 1.8CO + 0.2XO2 + 0.2RO2 + HO_2$ | (R116) | - |
| (R117) | $GLY + h\nu \rightarrow 2HO_2 + 2CO$ | (R117) | - | (R117) | - |
| (R118) | $GLY + NO_3 \rightarrow HNO_3 + CO + HO_2 + XO2 + RO2$ | (R118) | $GLY + NO_3 \rightarrow HNO_3 + 1.5CO + 0.5XO2 + 0.5RO2 + HO_2$ | (R118) | - |

| Reaction Number | CB6r1[a] | Reaction Number | CB6r2[b] | Reaction Number | CB6r3[c] |
|---|---|---|---|---|---|
| (R119) | $MGLY + h\nu \rightarrow C2O3 + HO_2 + CO$ | (R119) | - | (R119) | - |
| (R120) | $MGLY + NO_3 \rightarrow HNO_3 + C2O3 + XO2 + RO2$ | (R120) | - | (R120) | - |
| (R121) | $MGLY + OH \rightarrow C2O3 + CO$ | (R121) | - | (R121) | - |
| (R122) | $H_2 + OH \rightarrow HO_2$ | (R122) | - | (R122) | - |
| (R123) | $CO + OH \rightarrow HO_2$ | (R123) | - | (R123) | - |
| (R124) | $CH_4 + OH \rightarrow MEO2 + RO_2$ | (R124) | - | (R124) | - |
| (R125) | $ETHA + OH \rightarrow 0.991ALD2 + 0.991XO2H + 0.009XO2N + RO_2$ | (R125) | - | (R125) | - |
| (R126) | $MEOH + OH \rightarrow HCHO + HO_2$ | (R126) | - | (R126) | - |
| (R127) | $ETOH + OH \rightarrow 0.95ALD2 + 0.9HO_2 + 0.1XO2H + 0.1RO_2 + 0.078HCHO + 0.011GLYD$ | (R127) | - | (R127) | - |
| (R128) | $KET + h\nu \rightarrow 0.5ALD2 + 0.5C2O3 + 0.5XO2H + 0.5CXO3 + 0.5MEO2 + RO2 - 2.5PAR$ | (R128) | - | (R128) | - |
| (R129) | $ACET + h\nu \rightarrow 0.38CO + 1.38MEO2 + 1.38RO2 + 0.62C2O3$ | (R129) | - | (R129) | - |
| (R130) | $ACET + OH \rightarrow HCHO + C2O3 + XO2 + RO2$ | (R130) | - | (R130) | - |
| (R131) | $PRPA + OH \rightarrow 0.71ACET + 0.26ALDX + 0.26PAR + 0.97XO2H + 1.00RO2 + 0.03XO2N$ | (R131) | - | (R131) | $PRPA + OH \rightarrow XPRP$ |
| (R132) | $PAR + OH \rightarrow 0.11ALDX + 0.76ROR + 0.11XO2H + 0.76XO2 + RO2 - 0.11PAR + 0.13XO2N$ | (R132) | - | (R132) | $PAR + OH \rightarrow XPAR$ - |
| (R133) | $ROR \rightarrow 0.2KET + 0.42ACET + 0.74ALD2 + 0.37ALDX + 0.04XO2N + 0.94XO2H + 0.98RO2 + 0.02ROR - 2.7PAR$ | (R133) | - | (R133) | - |
| (R134) | $ROR + O_2 \rightarrow KET + HO_2$ | (R134) | - | (R134) | - |
| (R135) | $ROR + NO_2 \rightarrow NTR$ | (R135) | $ROR + NO_2 \rightarrow NTR2$ | (R135) | $ROR + NO_2 \rightarrow NTR1$ |
| (R136) | $ETHY + OH \rightarrow 0.7GLY + 0.7OH + 0.3FACD + 0.3CO + 0.3HO_2$ | (R136) | - | (R136) | - |
| (R137) | $ETH + O \rightarrow HCHO + HO_2 + CO + 0.7XO2H + 0.7RO2 + 0.3OH$ | (R137) | - | (R137) | - |
| (R138) | $ETH + OH \rightarrow XO2H + RO2 + 1.56HCHO + 0.22GLYD$ | (R138) | - | (R138) | - |
| (R139) | $ETH + O_3 \rightarrow HCHO + 0.51CO + 0.16HO_2 + 0.16OH + 0.37FACD$ | (R139) | - | (R139) | - |
| (R140) | $ETH + NO_3 \rightarrow 0.5NO_2 + 0.5NTR + 0.5XO2H + 0.5XO2 + RO2 + 1.12HCHO$ | (R140) | $ETH + NO_3 \rightarrow 0.5NO_2 + 0.5NTR1 + 0.5XO2H + 0.5XO2 + RO2 + 1.12HCHO$ | (R140) | - |
| (R141) | $OLE + O \rightarrow 0.2ALD2 + 0.3ALDX + 0.1HO_2 + 0.2XO2H + 0.2CO + 0.2HCHO + 0.01XO2N + 0.21RO2 + 0.2PAR + 0.1OH$ | (R141) | - | (R141) | - |

| Reaction Number | CB6r1[a] | Reaction Number | CB6r2[b] | Reaction Number | CB6r3[c] |
|---|---|---|---|---|---|
| (R142) | $OLE + OH \rightarrow 0.78HCHO + 0.49ALD2 + 0.49ALDX + 0.98XO2H + 0.2XO2 + 0.02XO2N + 1.2RO2 - 0.73PAR$ | (R142) | - | (R142) | - |
| (R143) | $OLE + O_3 \rightarrow 0.29ALD2 + 0.56HCHO + 0.27ALDX + 0.15XO2H + 0.15RO2 + 0.33OH + 0.08HO_2 + 0.38CO + 0.07GLY + 0.07MGLY + 0.09FACD + 0.13AACD + 0.04H_2O_2 - 0.79PAR$ | (R143) | - | (R143) | - |
| (R144) | $OLE + NO_3 \rightarrow 0.5NO_2 + 0.5NTR + 0.48XO2 + 0.48XO2H + 0.04XO2N + RO2 + 0.5HCHO + 0.25ALD2 + 0.38ALDX - PAR$ | (R144) | $OLE + NO_3 \rightarrow 0.5NO_2 + 0.5NTR1 + 0.48XO2 + 0.48XO2H + 0.04XO2N + RO2 + 0.5HCHO + 0.25ALD2 + 0.38ALDX - PAR$ | (R144) | - |
| (R145) | $IOLE + O \rightarrow 1.24ALD2 + 0.66ALDX + 0.1XO2H + 0.1RO2 + 0.1CO + 0.1PAR$ | (R145) | - | (R145) | - |
| (R146) | $IOLE + OH \rightarrow 1.30ALD2 + 0.7ALDX + XO2H + RO2$ | (R146) | - | (R146) | - |
| (R147) | $IOLE + O_3 \rightarrow 0.73ALD2 + 0.44ALDX + 0.13HCHO + 0.24CO + 0.5OH + 0.3XO2H + 0.3RO2 + 0.24GLY + 0.06MGLY + 0.29PAR + 0.08AACD + 0.08H_2O_2$ | (R147) | - | (R147) | - |
| (R148) | $IOLE + NO_3 \rightarrow 0.5NO_2 + 0.5NTR + 0.48XO2 + 0.48XO2H + 0.04XO2N + RO2 + 0.5ALD2 + 0.62ALDX + PAR$ | (R148) | $IOLE + NO_3 \rightarrow 0.5NO_2 + 0.5NTR1 + 0.48XO2 + 0.48XO2H + 0.04XO2N + RO2 + 0.5ALD2 + 0.62ALDX + PAR$ | (R148) | - |
| (R149) | $ISOP + OH \rightarrow ISO2RO2+$ | (R149) | - | (R149) | - |
| (R150) | | (R150) | $ISOP + O \rightarrow 0.75ISPD + 0.5HCHO + 0.25XO2 + 0.25RO2 + 0.25HO_2 + 0.25CXO3 + 0.25PAR$ | (R150) | - |
| (R151) | $ISO2 + NO \rightarrow 0.12INTR + 0.88NO_2 + 0.8HO_2 + 0.66HCHO + 0.66ISPD + 0.08XO2H + 0.08RO2 + 0.05IOLE + 0.04GLYD + 0.12PAR + 0.04GLY + 0.04MGLY + 0.09OLE + 0.12ALDX$ | (R151) | $ISO2 + NO \rightarrow 0.1INTR + 0.9NO_2 + 0.67HCHO + 0.9ISPD + 0.82HO_2 + 0.08XO2H + 0.08RO2$ | (R151) | - |
| (R152) | $ISO2 + HO_2 \rightarrow 0.88ISPX + 0.12OH + 0.12HO_2 + 0.12HCHO + 0.12ISPD$ | (R152) | - | (R152) | - |
| (R153) | $ISO2 + C2O3 \rightarrow 0.71HO_2 + 0.58HCHO + 0.58ISPD + 0.07XO2H + 0.04IOLE + 0.04GLYD + 0.1PAR + 0.03GLY + 0.04MGLY + 0.08OLE + 0.1ALDX + 0.8MEO2 + 0.2AACD + 0.87RO2$ | (R153) | $ISO2 + C2O3 \rightarrow 0.6HCHO + ISPD + 0.73HO_2 + 0.07XO2H + 0.8MEO2 + 0.2AACD + 0.87RO2$ | (R153) | - |
| (R154) | $ISO2 + RO2 \rightarrow 0.8HO_2 + 0.66HCHO + 0.66ISPD + 0.08XO2H + 0.05IOLE + 0.04GLYD + 0.12PAR + 0.04GLY + 0.04MGLY + 0.09OLE + 0.12ALDX + 1.08RO2$ | (R154) | $ISO2 + RO2 \rightarrow 0.6HCHO + ISPD + 0.73HO_2 + 0.07XO2H + 1.07RO2$ | (R154) | - |

| Reaction Number | CB6r1[a] | Reaction Number | CB6r2[b] | Reaction Number | CB6r3[c] |
|---|---|---|---|---|---|
| (R155) | $ISO2 \rightarrow 0.8HO_2 + 0.04OH + 0.04HCHO + 0.8ISPD$ | (R155) | $ISO2 \rightarrow HO_2 + HPLD$ | (R155) | - |
| (R156) | $ISOP + O_3 \rightarrow 0.6HCHO + 0.65ISPD + 0.15ALDX + 0.2CXO3 + 0.35PAR + 0.27OH + 0.2XO2 + 0.2RO2 + 0.07HO_2 + 0.07CO$ | (R156) | - | (R156) | - |
| (R157) | $ISOP + NO_3 \rightarrow 0.35NO_2 + 0.65INTR + 0.64XO2H + 0.33XO2 + 0.03XO2N + RO2 + 0.35HCHO + 0.35ISPD$ | (R157) | $ISOP + NO_3 \rightarrow 0.35NO_2 + 0.65NTR2 + 0.64XO2H + 0.33XO2 + 0.03XO2N + RO2 + 0.35HCHO + 0.35ISPD$ | (R157) | - |
| (R158) | $ISPD + OH \rightarrow 0.1XO2N + 0.38XO2 + 0.32XO2H + 0.79RO2 + 0.84PAR + 0.38C2O3 + 0.21CXO3 + 0.38GLYD + 0.24MGLY + 0.24HCHO + 0.07OLE + 0.08CO + 0.03ALDX$ | (R158) | $ISPD + OH \rightarrow 0.06XO2N + 0.52XO2 + 0.24XO2H + 0.15MGLY + 0.27MEO2 + 0.12GLY + 0.35GLYD + 0.23C2O3 + 0.12CXO3 + 0.24PAR + 0.26ACET + 0.2CO + 0.14HO_2 + 1.09RO2$ | (R158) | $ISPD + OH \rightarrow 0.02XO2N + 0.52XO2 + 0.12MGLY + 0.12MEO2 + 0.27GLYD + 0.27C2O3 + 0.46OPO3 + 0.12PAR + 0.14ACET + 0.14CO + 0.14HO_2 + 0.66RO2$ |
| (R159) | $ISPD + O_3 \rightarrow 0.02ALD2 + 0.15HCHO + 0.23CO + 0.85MGLY + 0.36PAR + 0.11C2O3 + 0.06XO2H + 0.06RO2 + 0.27OH + 0.09HO_2$ | (R159) | $ISPD + O_3 \rightarrow 0.04ALD2 + 0.23HCHO + 0.53MGLY + 0.17GLY + 0.17ACET + 0.54CO + 0.46OH + 0.15FACD + 0.4HO_2 + 0.14C2O3$ | (R159) | - |
| (R160) | $ISPD + NO_3 \rightarrow 0.64CO + 0.28HCHO + 0.36ALDX + 1.28PAR + 0.85HO_2 + 0.07CXO3 + 0.07XO2H + 0.07RO2 + 0.85NTR + 0.15HNO_3$ | (R160) | $ISPD + NO_3 \rightarrow 0.72HNO_3 + 0.14NTR2 + 0.14NO_2 + 0.14XO2 + 0.14XO2H + 0.11GLYD + 0.11MGLY + 0.72PAR + 0.72CXO3 + 0.28RO2$ | (R160) | - |
| (R161) | $ISPD + h\nu \rightarrow 0.33CO + 0.07ALD2 + 0.9HCHO + 0.83PAR + 0.33HO_2 + 0.7XO2H + 0.7RO2 + 0.97C2O3$ | (R161) | $ISPD + h\nu \rightarrow 0.76HO_2 + 0.34XO2H + 0.16XO2 + 0.34MEO2 + 0.21C2O3 + 0.26HCHO + 0.24OLE + 0.24PAR + 0.17ACET + 0.13GLYD + 0.84RO2$ | (R161) | - |
| (R162) | $ISPX + OH \rightarrow 0.9EPOX + 0.93OH + 0.07ISO2 + 0.07RO2 + 0.03IOLE + 0.03ALDX$ | (R162) | - | (R162) | - |
| (R163) | | (R163) | $HPLD \rightarrow OH + ISPD + HO_2$ | (R163) | $HPLD \rightarrow OH + ISPD$ |
| (R164) | | (R164) | $HPLD + NO_3 \rightarrow HNO_3 + ISPD$ | (R164) | - |
| (R165) | $EPOX + OH \rightarrow EPX2 + RO2$ | (R165) | - | (R165) | - |
| (R166) | $EPX2 + HO_2 \rightarrow 0.28GLYD + 0.28GLY + 0.28MGLY + 1.12OH + 0.82HO_2 + 0.38HCHO + 0.07FACD + 0.25CO + 2.17PAR$ | (R166) | - | (R166) | - |
| (R167) | $EPX2 + NO \rightarrow 0.28GLYD + 0.28GLY + 0.28MGLY + 0.12OH + 0.82HO_2 + 0.38HCHO + NO_2 + 0.25CO + 2.17PAR$ | (R167) | - | (R167) | - |
| (R168) | $EPX2 + C2O3 \rightarrow 0.22GLYD + 0.22GLY + 0.22MGLY + 0.1OH + 0.66HO_2 + 0.3HCHO + 0.2CO + 1.74PAR + 0.8MEO2 + 0.2AACD + 0.8RO2$ | (R168) | - | (R168) | - |
| (R169) | $EPX2 + RO2 \rightarrow 0.28GLYD + 0.28GLY + 0.28MGLY + 0.12OH + 0.82HO_2 + 0.38HCHO + 0.25CO + 2.17PAR + RO2$ | (R169) | - | (R169) | - |
| (R170) | $INTR + OH \rightarrow 0.63XO2 + 0.37XO2H + RO2 + 0.44NO_2 + 0.18NO_3 + 0.1INTR + 0.59HCHO + 0.33GLYD + 0.18FACD + 2.70PAR + 0.1OLE + 0.08ALDX + 0.27NTR$ | (R170) | $INTR + OH \rightarrow 0.63XO2 + 0.37XO2H + RO2 + 0.44NO_2 + 0.18NO_3 + 0.1INTR + 0.59HCHO + 0.33GLYD + 0.18FACD + 2.70PAR + 0.1OLE + 0.08ALDX + 0.27NTR2$ | (R170) | - |

| Reaction Number | CB6r1[a] | Reaction Number | CB6r2[b] | Reaction Number | CB6r3[c] |
|---|---|---|---|---|---|
| (R171) | TERP + O → 0.15ALDX + 5.12PAR | (R171) | - | (R171) | - |
| (R172) | TERP + OH → 0.75XO2H + 0.5XO2+ 0.25XO2N + 1.5RO2 + 0.28HCHO + 1.66PAR+ 0.47ALDX | (R172) | - | (R172) | - |
| (R173) | TERP + $O_3$ → 0.57OH + 0.07XO2H+ 0.69XO2 + 0.18XO2N + 0.94RO2 + 0.24HCHO+ 0.001CO + 7PAR + 0.21ALDX + 0.39CXO3 | (R173) | - | (R173) | - |
| (R174) | TERP + $NO_3$ → 0.47$NO_2$ + 0.28XO2H+ 0.75XO2 + 0.25XO2N + 1.28RO2 + 0.47ALDX+ 0.53NTR | (R174) | TERP + $NO_3$ → 0.47$NO_2$ + 0.28XO2H+ 0.75XO2 + 0.25XO2N + 1.28RO2 + 0.47ALDX+ 0.53NTR2 | (R174) | - |
| (R175) | BENZ + OH → 0.53CRES + 0.35BZO2+ 0.35RO2 + 0.12OPEN + 0.12OH + 0.53$HO_2$ | (R175) | - | (R175) | - |
| (R176) | BZO2 + NO → 0.92$NO_2$ + 0.08NTR+ 0.92GLY + 0.92OPEN + 0.92$HO_2$ | (R176) | BZO2 + NO → 0.92$NO_2$ + 0.08NTR2+ 0.92GLY + 0.92OPEN + 0.92$HO_2$ | (R176) | - |
| (R177) | BZO2 + C2O3 → GLY + OPEN+ $HO_2$ + MEO2 + RO2 | (R177) | - | (R177) | - |
| (R178) | BZO2 + $HO_2$ → | (R178) | - | (R178) | - |
| (R179) | BZO2 + RO2 → GLY + OPEN+ $HO_2$ + RO2 | (R179) | - | (R179) | - |
| (R180) | TOL + OH → 0.18CRES + 0.65TO2+ 0.72RO2 + 0.1OPEN + 0.1OH + 0.07XO2H+ 0.18$HO_2$ | (R180) | - | (R180) | - |
| (R181) | TO2 + NO → 0.86$NO_2$ + 0.14NTR+ 0.42GLY + 0.44MGLY + 0.66OPEN + 0.2XOPN+ 0.86$HO_2$ | (R181) | TO2 + NO → 0.86$NO_2$ + 0.14NTR2+ 0.42GLY + 0.44MGLY + 0.66OPEN + 0.2XOPN+ 0.86$HO_2$ | (R181) | - |
| (R182) | TO2 + C2O3 → 0.48GLY + 0.52MGLY+ 0.77OPEN + 0.23XOPN + $HO_2$ + MEO2+ RO2 | (R182) | - | (R182) | - |
| (R183) | TO2 + $HO_2$ → | (R183) | - | (R183) | - |
| (R184) | TO2 + RO2 → 0.48GLY + 0.52MGLY+ 0.77OPEN + 0.23XOPN + $HO_2$ + RO2 | (R184) | - | (R184) | - |
| (R185) | XYL + OH → 0.15CRES + 0.54XLO2+ 0.6RO2 + 0.24XOPN + 0.24OH + 0.06XO2H+ 0.15$HO_2$ | (R185) | - | (R185) | - |

| Reaction Number | CB6r1[a] | Reaction Number | CB6r2[b] | Reaction Number | CB6r3[c] |
|---|---|---|---|---|---|
| (R186) | $XLO2 + NO \rightarrow 0.86NO_2 + 0.14NTR + 0.22GLY + 0.68MGLY + 0.3OPEN + 0.56XOPN + 0.86HO_2$ | (R186) | $XLO2 + NO \rightarrow 0.86NO_2 + 0.14NTR2 + 0.22GLY + 0.68MGLY + 0.3OPEN + 0.56XOPN + 0.86HO_2$ | (R186) | - |
| (R187) | $XLO2 + HO_2 \rightarrow$ | (R187) | - | (R187) | - |
| (R188) | $XLO2 + C2O3 \rightarrow 0.26GLY + 0.77MGLY + 0.35OPEN + 0.65XOPN + HO_2 + MEO2 + RO2$ | (R188) | - | (R188) | - |
| (R189) | $XLO2 + RO2 \rightarrow 0.26GLY + 0.77MGLY + 0.35OPEN + 0.65XOPN + HO_2 + RO_2$ | (R189) | - | (R189) | - |
| (R190) | $CRES + OH \rightarrow 0.06CRO + 0.12XO2H + HO_2 + 0.13OPEN + 0.73CAT1 + 0.06CO + 0.06XO2N + 0.18RO2 + 0.06HCHO$ | (R190) | $CRES + OH \rightarrow 0.03GLY + 0.03OPEN + HO_2 + 0.2CRO + 0.73CAT1 + 0.02XO2N + 0.02RO2$ | (R190) | - |
| (R191) | $CRES + NO_3 \rightarrow 0.30CRO + HNO_3 + 0.24XO2 + 0.36XO2H + 0.48ALDX + 0.24HCHO + 0.24MGLY + 0.12OPEN + 0.1XO2N + 0.7RO2 + 0.24CO$ | (R191) | $CRES + NO_3 \rightarrow 0.3CRO + HNO_3 + 0.48XO2 + 0.12XO2H + 0.24GLY + 0.24MGLY + 0.48OPO3 + 0.1XO2N + 0.7RO2$ | (R191) | - |
| (R192) | $CRO + NO_2 \rightarrow CRON$ | (R192) | - | (R192) | - |
| (R193) | $CRO + HO_2 \rightarrow CRES$ | (R193) | - | (R193) | - |
| (R194) | $CRON + OH \rightarrow CRNO$ | (R194) | $CRON + OH \rightarrow NTR2 + 0.5CRO$ | (R194) | - |
| (R195) | $CRON + NO_3 \rightarrow CRNO + HNO_3$ | (R195) | $CRON + NO_3 \rightarrow NTR2 + 0.5CRO + HNO_3$ | (R195) | - |
| (R196) | | (R196) | $CRON + h\nu \rightarrow HONO + HO_2 + HCHO + OPEN$ | (R196) | - |
| (R197) | $XOPN + h\nu \rightarrow CAO2 + 0.7HO_2 + 0.7CO + 0.3C2O3 + RO2$ | (R197) | $XOPN + h\nu \rightarrow 0.4GLY + XO2H + 0.7HO_2 + 0.7CO + 0.3C2O3$ | (R197) | - |
| (R198) | $XOPN + OH \rightarrow MGLY + CAO2 + XO2H + RO2$ | (R198) | $XOPN + OH \rightarrow MGLY + 0.4GLY + 2XO2H + 2RO2$ | (R198) | - |
| (R199) | $XOPN + O_3 \rightarrow 1.2MGLY + 0.5OH + 0.6C2O3 + 0.1ALD2 + 0.5CO + 0.3XO2H + 0.3RO2$ | (R199) | - | (R199) | - |
| (R200) | $XOPN + NO_3 \rightarrow 0.5NO_2 + 0.5NTR + 0.45XO2H + 0.45XO2 + 0.1XO2N + RO2 + 0.25OPEN + 0.25MGLY$ | (R200) | $XOPN + NO_3 \rightarrow 0.5NO_2 + 0.5NTR2 + 0.45XO2H + 0.45XO2 + 0.1XO2N + RO2 + 0.25OPEN + 0.25MGLY$ | (R200) | - |
| (R201) | $OPEN + h\nu \rightarrow OPO3 + HO_2 + CO$ | (R201) | - | (R201) | - |
| (R202) | $OPEN + OH \rightarrow 0.6OPO3 + 0.4RO2 + 0.4CAO2$ | (R202) | $OPEN + OH \rightarrow 0.6OPO3 + 0.4XO2H + 0.4RO2 + 0.4GLY$ | (R202) | - |
| (R203) | $OPEN + O_3 \rightarrow 1.4GLY + 0.24MGLY + 0.5OH + 0.12C2O3 + 0.08HCHO + 0.02ALD2 + 1.98CO + 0.56HO_2$ | (R203) | - | (R203) | - |
| (R204) | $OPEN + NO_3 \rightarrow OPO3 + HNO_3$ | (R204) | - | (R204) | - |

| Reaction Number | CB6r1[a] | Reaction Number | CB6r2[b] | Reaction Number | CB6r3[c] |
|---|---|---|---|---|---|
| (R205) | $CAT1 + OH \rightarrow CAO2 + RO2$ | (R205) | $CAT1 + OH \rightarrow 0.14HCHO + 0.2HO_2 + 0.5CRO$ | (R205) | - |
| (R206) | $CAT1 + NO_3 \rightarrow CRO + HNO_3$ | (R206) | - | (R206) | - |
| (R207) | $OPO3 + NO \rightarrow NO_2 + XO2H + RO2 + ALDX$ | (R207) | $OPO3 + NO \rightarrow NO_2 + 0.5GLY + 0.5CO + 0.8HO_2 + 0.2CXO3$ | (R207) | - |
| (R208) | $OPO3 + NO_2 \rightarrow OPAN$ | (R208) | - | (R208) | - |
| (R209) | $OPAN \rightarrow OPO3 + NO_2$ | (R209) | - | (R209) | - |
| (R210) | $OPO3 + HO_2 \rightarrow 0.41PACD + 0.15AACD + 0.15O_3 + 0.44ALDX + 0.44XO2H + 0.44RO2 + 0.44OH$ | (R210) | - | (R210) | - |
| (R211) | $OPO3 + C2O3 \rightarrow MEO2 + XO2 + ALDX + 2RO2$ | (R211) | - | (R211) | - |
| (R212) | $OPO3 + RO2 \rightarrow 0.8XO2H + 0.8ALDX + 1.8RO2 + 0.2AACD$ | (R212) | - | (R212) | - |
| | | (R213) | $OPAN + OH \rightarrow 0.5NO_2 + 0.5GLY + CO + 0.5NTR2$ | (R213) | - |
| | | (R214) | $PANX + OH \rightarrow ALD2 + NO_2$ | (R213) | - |
| | | | | (R214) | - |
| | | | | (R215) | $NAPH + OH \rightarrow 0.15CRES + 0.54XLO2 + 0.6RO2 + 0.24XOPN + 0.24OH + 0.06XO2H + 0.15HO_2$ |
| | | | | (R216) | $ECH4 + OH \rightarrow MEO2 + RO2$ |
| | | | | (R217) | $XPRP \rightarrow XO2N + RO2$ |
| | | | | (R218) | $XPRP \rightarrow 0.73ACET + 0.27ALDX + 0.27PAR + XO2H + RO2$ |
| | | | | (R219) | $XPAR \rightarrow XO2N + RO2$ |
| | | | | (R220) | $XPAR \rightarrow 0.13ALDX + 0.87ROR + 0.13XO2H + 0.87XO2 + RO2 - 0.13PAR$ |
| | | (R215) | $NTR2 \rightarrow HNO_3$ | (R221) | - |
| (R213) | $CRNO + NO_2 \rightarrow 2NTR$ | | | | |
| (R214) | $CRNO + O_3 \rightarrow CRN2$ | | | | |
| (R215) | $CRN2 + NO \rightarrow CRNO + NO_2$ | | | | |
| (R216) | $CRN2 + HO_2 \rightarrow CRPX$ | | | | |
| (R217) | $CRPX + h\nu \rightarrow CRNO + OH$ | | | | |
| (R218) | $CRPX + OH \rightarrow CRN2$ | | | | |
| (R219) | $CAO2 + NO \rightarrow 0.86NO_2 + 0.14NTR + 1.2HO_2 + 0.34HCHO + 0.34CO$ | | | | |
| (R220) | $CAO2 + HO_2 \rightarrow$ | | | | |
| (R221) | $CAO2 + C2O3 \rightarrow HO_2 + 0.4GLY + MEO2 + RO2$ | | | | |
| (R222) | $CAO2 + RO2 \rightarrow HO_2 + 0.4GLY + RO2$ | | | | |

| Reaction Number | CB6r1[a] | Reaction Number | CB6r2[b] | Reaction Number | CB6r3[c] |
|---|---|---|---|---|---|
| (R223)[d] | $\rightarrow$ NO | (R216) | - | (R222) | - |
| (R224)[d] | $\rightarrow$ NO$_2$ | (R217) | - | (R223) | - |
| (R225)[d] | $\rightarrow$ CO | (R218) | - | (R224) | - |
| (R226)[d] | $\rightarrow$ HCHO | (R219) | - | (R225) | - |
| (R227)[d] | $\rightarrow$ ALD2 | (R220) | - | (R226) | - |
| (R228)[d] | $\rightarrow$ IOLE | (R221) | - | (R227) | - |
| (R229)[d] | $\rightarrow$ ALDX | (R222) | - | (R228) | - |
| (R230)[d] | $\rightarrow$ ETH | (R223) | - | (R229) | - |
| (R231)[d] | $\rightarrow$ TOL | (R224) | - | (R230) | - |
| (R232)[d] | $\rightarrow$ XYL | (R225) | - | (R231) | - |
| (R233)[d] | $\rightarrow$ ISOP | (R226) | - | (R232) | - |
| (R234)[d] | $\rightarrow$ PAR | (R227) | - | (R233) | - |
| (R235)[d] | $\rightarrow$ OLE | (R228) | - | (R234) | - |
| (R236)[e] | O$_3$ $\rightarrow$ | (R229) | - | (R235) | - |
| (R237)[e] | NO $\rightarrow$ | (R230) | - | (R236) | - |
| (R238)[e] | NO$_2$ $\rightarrow$ | (R231) | - | (R237) | - |
| (R239)[e] | HNO$_3$ $\rightarrow$ | (R232) | - | (R238) | - |
| (R240)[e] | H$_2$O$_2$ $\rightarrow$ | (R233) | - | (R239) | - |
| (R241)[e] | N$_2$O$_5$ $\rightarrow$ | (R234) | - | (R240) | - |
| (R242)[e] | CO $\rightarrow$ | (R235) | - | (R241) | - |
| (R243)[e] | HONO $\rightarrow$ | (R236) | - | (R242) | - |
| (R244)[e] | HCHO $\rightarrow$ | (R237) | - | (R243) | - |

Notes: [a] Yarwood et al. (2012).

[b] Ruiz Hildebrandt and Yarwood (2013).

[c] Emery et al. (2015).

[d] Added to the mechanism to represent surface emissions, not included in the original mechanism.

[e] Added to the mechanism to represent dry depositions, not included in the original mechanism.

*Acknowledgements.* This work was supported by the National Key R&D Program of China (Grant No. 2017YFC0209801) and the National Natural Science Foundation of China (Grant No. 41375044). The numerical calculations in this paper have been done on the high performance

computing system in the High Performance Computing Center, Nanjing University of Information Science & Technology. The authors also like to thank Rolf Sander from Max-Planck Institute for Chemistry for his valuable suggestions on the improvements of the model.

*Author contributions.* Le Cao conceived the idea of the article and wrote the python script for converting the data format. Le Cao also configured and performed the computations. Simeng Li revised the chemical mechanisms and wrote the paper with Le Cao together. Luhang Sun revised the paper and gave valuable suggestions. All the authors listed have read and approved the final manuscript.

*Competing interests.* The authors declare no conflict of interest.

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

**Table 1.** The initial air composition and the surface emission intensities used in two different simulation scenarios. This initial condition was adapted from Saylor and Ford (1995) and Sandu et al. (1997), and represents a heavily polluted environment with a 70% relative humidity.

| Species | Initial Concentration (ppb) | Weak Emission (ppb h$^{-1}$) | Strong Emission (ppb h$^{-1}$) |
|---------|------------------------------|------------------------------|--------------------------------|
| NO | 5 | 0.01 | 0.25 |
| NO$_2$ | 2 | 0.01 | 0.05 |
| HONO | 1 | - | - |
| O$_3$ | 100 | - | - |
| CO | 300 | - | 2.00 |
| HCHO | 10 | - | 0.20 |
| ALD2 | 2.2 | - | 0.04 |
| IOLE | 6.7 | - | 0.13 |
| ALDX | 1.1 | - | 0.02 |
| PAN | 1 | - | - |
| ETH | 10 | - | 0.20 |
| TOL | 10 | - | 0.20 |
| XYL | 10 | - | 0.20 |
| ISOP | 10 | 0.10 | 1.00 |
| PAR | 50 | - | 2.00 |
| OLE | 10 | - | 1.00 |
| H$_2$ | 560 | - | - |
| CH$_4$ | 1850 | - | - |
| H$_2$O | $2.17 \times 10^7$ | - | - |

**Table 2.** Dry deposition velocities used in model simulations for different atmospheric constituents.

| Species | Deposition Velocity $(\text{cm}\,\text{s}^{-1})$ | Reference |
|---------|---------------------------------------------------|-----------|
| $O_3$ | 0.4 | Seinfeld and Pandis (2006) |
| NO | 0.016 | Seinfeld and Pandis (2006) |
| $NO_2$ | 0.1 | Seinfeld and Pandis (2006) |
| $HNO_3$ | 4.0 | Seinfeld and Pandis (2006) |
| $H_2O_2$ | 0.5 | Seinfeld and Pandis (2006) |
| $N_2O_5$ | 4.0 | Hauglustaine et al. (1994) |
| CO | 0.03 | Seinfeld and Pandis (2006) |
| HONO | 4.0 | Hauglustaine et al. (1994) |
| HCHO | 6.0 | Seyfioglu et al. (2006) |

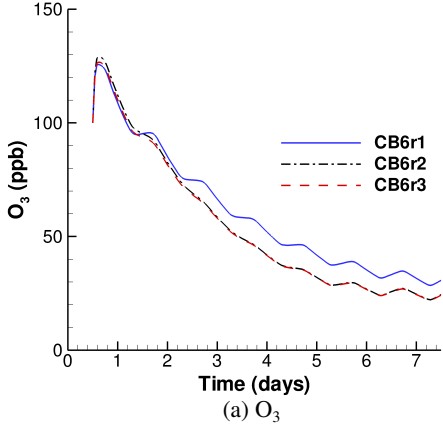

(a) $O_3$

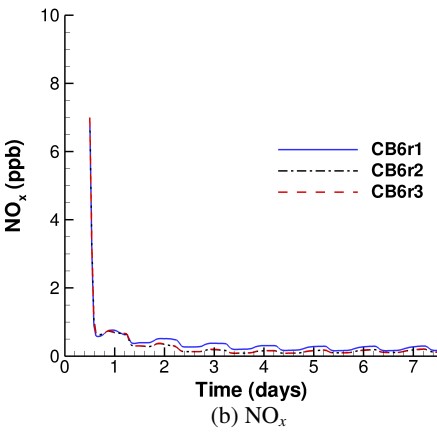

(b) $NO_x$

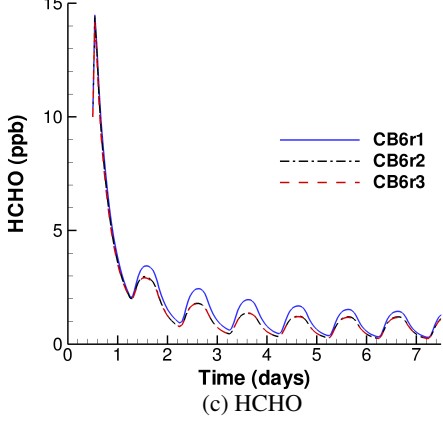

(c) HCHO

**Figure 1.** Simulated temporal evolution of $O_3$, $NO_x$ and HCHO by using different versions of the CB6 mechanism, when the surface emission is weak.

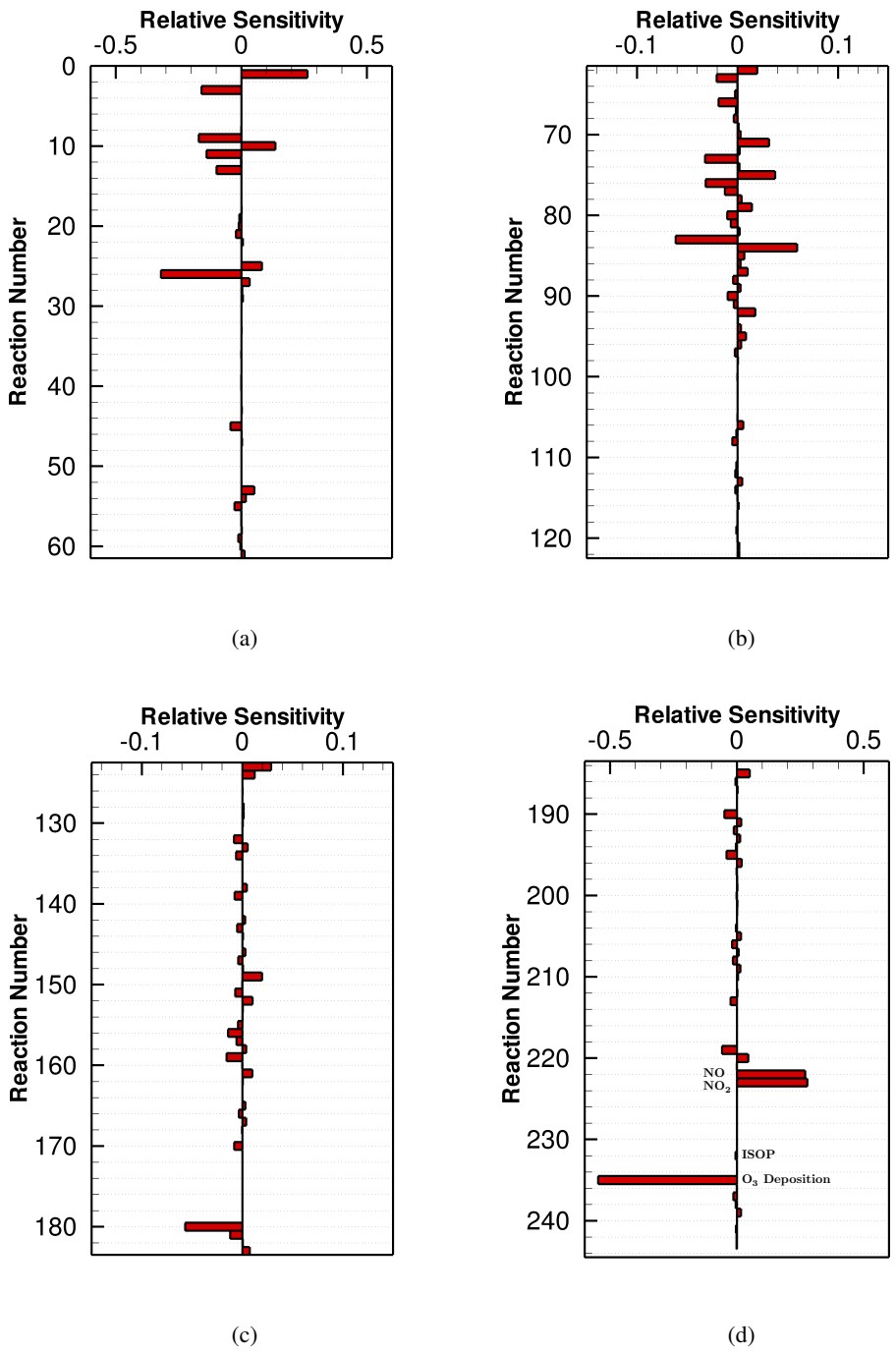

**Figure 2.** Averaged ozone sensitivity to the CB6r3 mechanism over the 7-th day, when the surface emission is weak. Note that the horizontal scales of the sub-figures are different. All the values of the sensitivities shown in this figure can be found in Tab. S1 of the supplementary material.

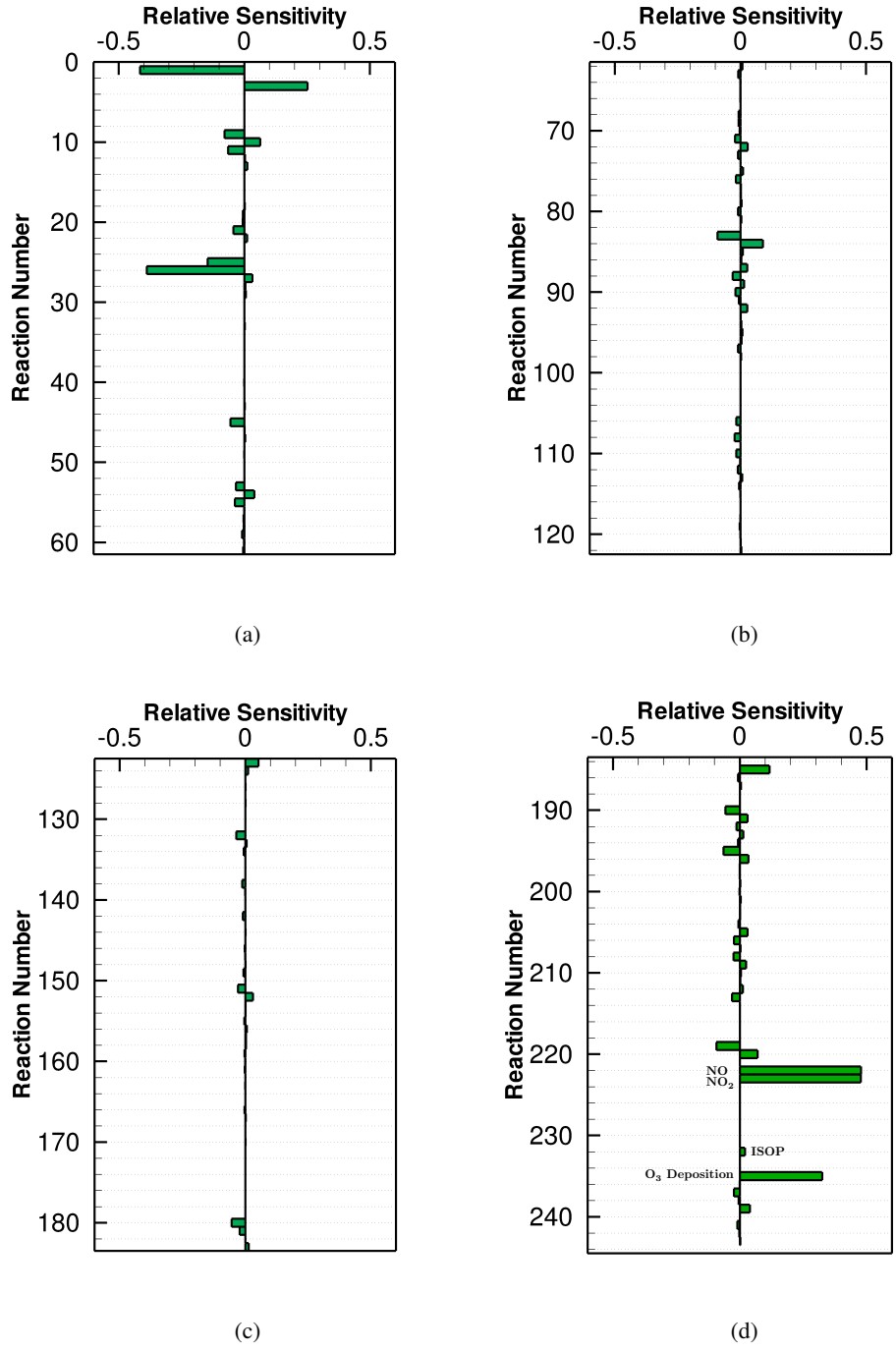

**Figure 3.** Averaged NO$_x$ sensitivity to the CB6r3 mechanism over the 7-th day, when the surface emission is weak. Note that the horizontal scales of the sub-figures are different. All the values of the sensitivities shown in this figure can be found in Tab. S1 of the supplementary material.

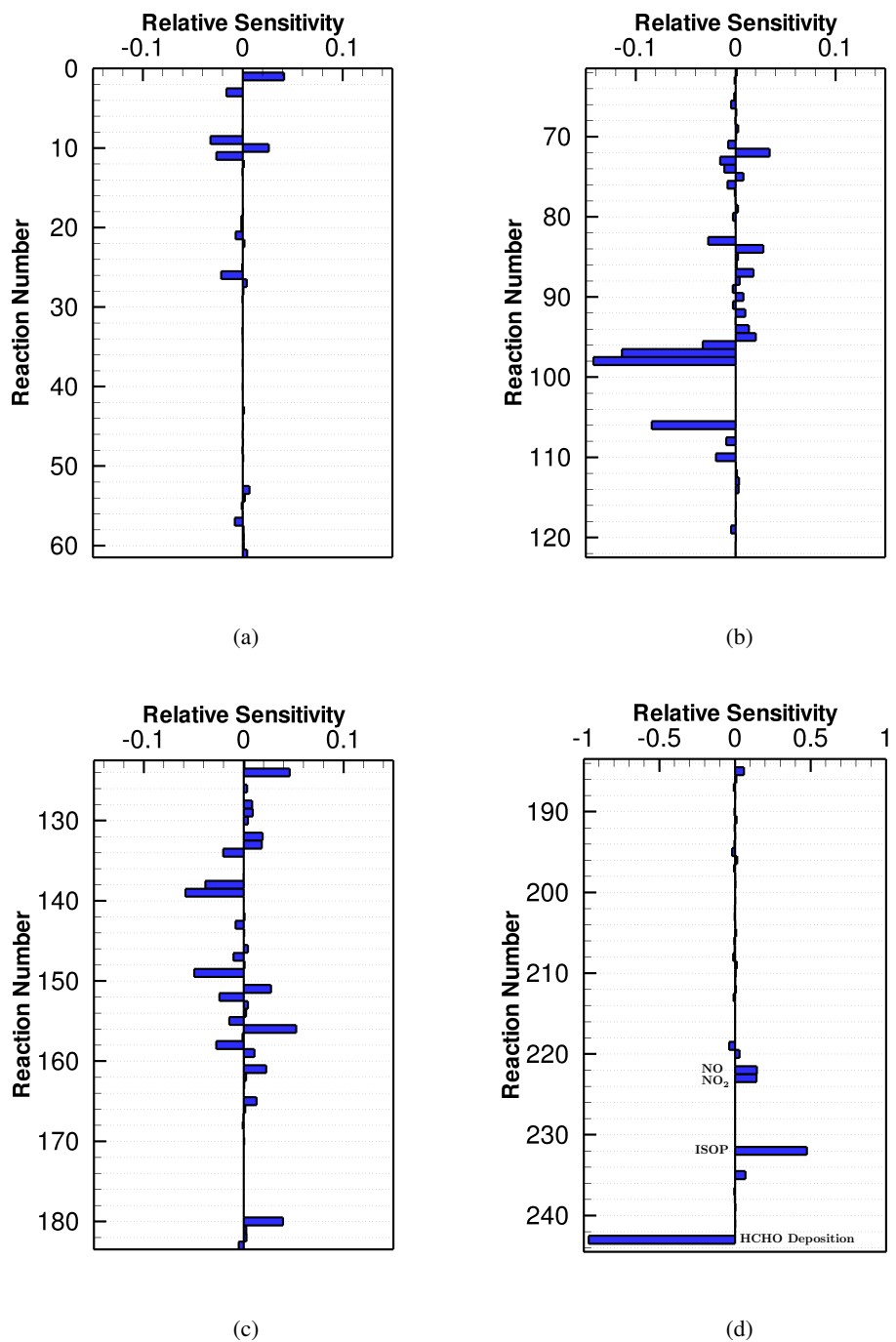

**Figure 4.** Averaged HCHO sensitivity to the CB6r3 mechanism over the 7-th day, when the surface emission is weak. Note that the horizontal scales of the sub-figures are different. All the values of the sensitivities shown in this figure can be found in Tab. S1 of the supplementary material.

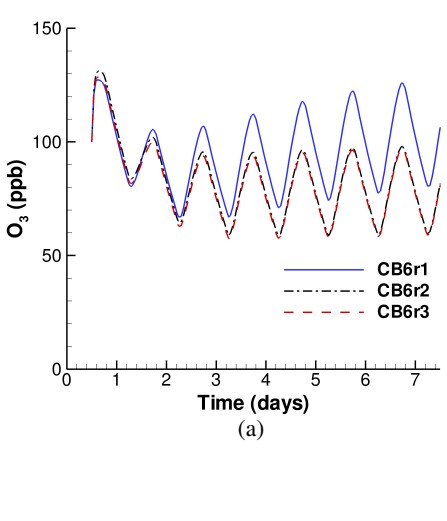

(a)

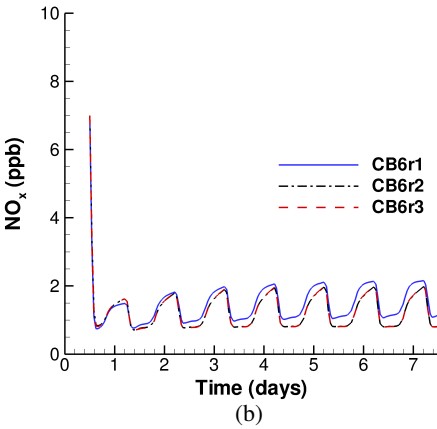

(b)

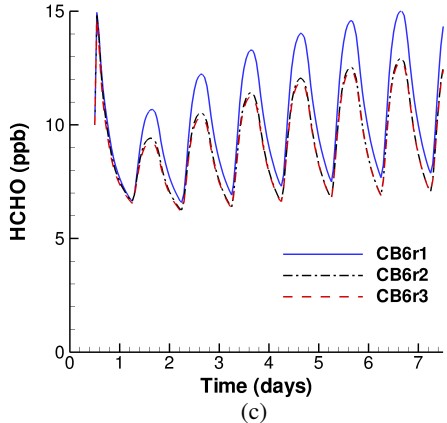

(c)

**Figure 5.** Simulated temporal evolution of $O_3$, $NO_x$ and HCHO by using different versions of the CB6 mechanism, when the surface emission is strong.

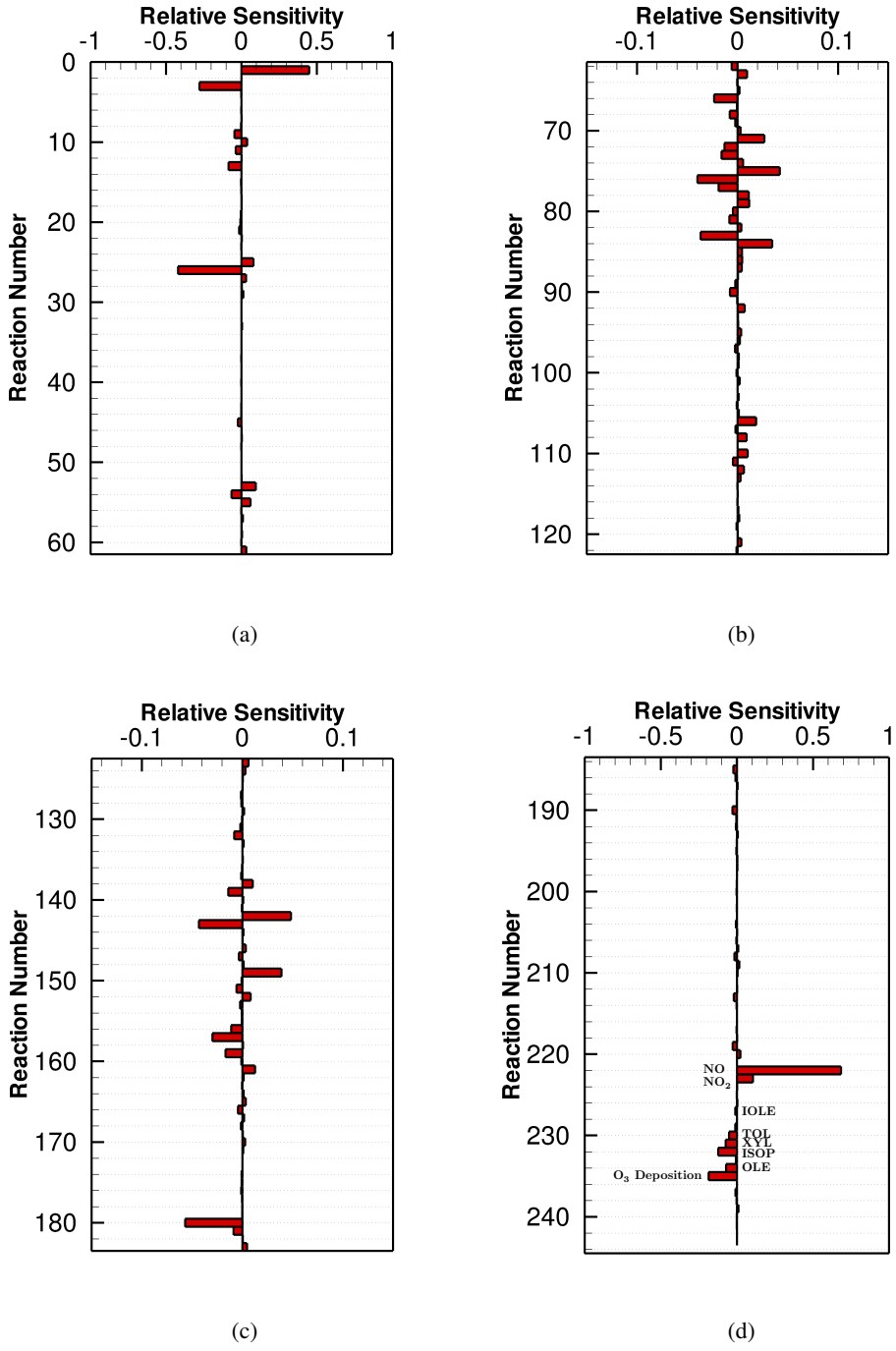

**Figure 6.** Averaged ozone sensitivity to the CB6r3 mechanism over the 7-th day, when the surface emission is strong. All the values of the sensitivities shown in this figure can be found in Tab. S1 of the supplementary material.

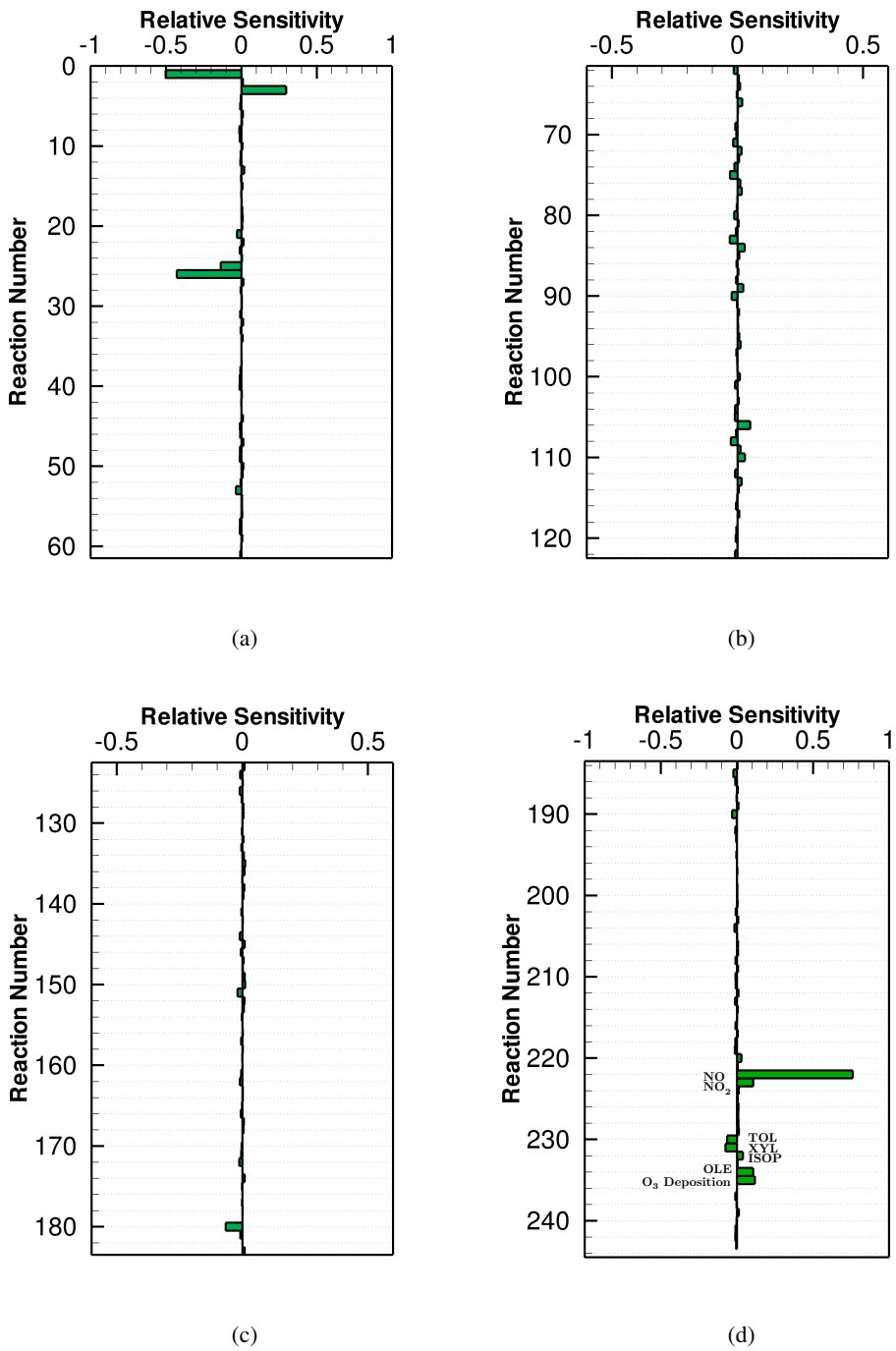

**Figure 7.** Averaged NO$_x$ sensitivity to the CB6r3 mechanism over the 7-th day, when the surface emission is strong. All the values of the sensitivities shown in this figure can be found in Tab. S1 of the supplementary material.

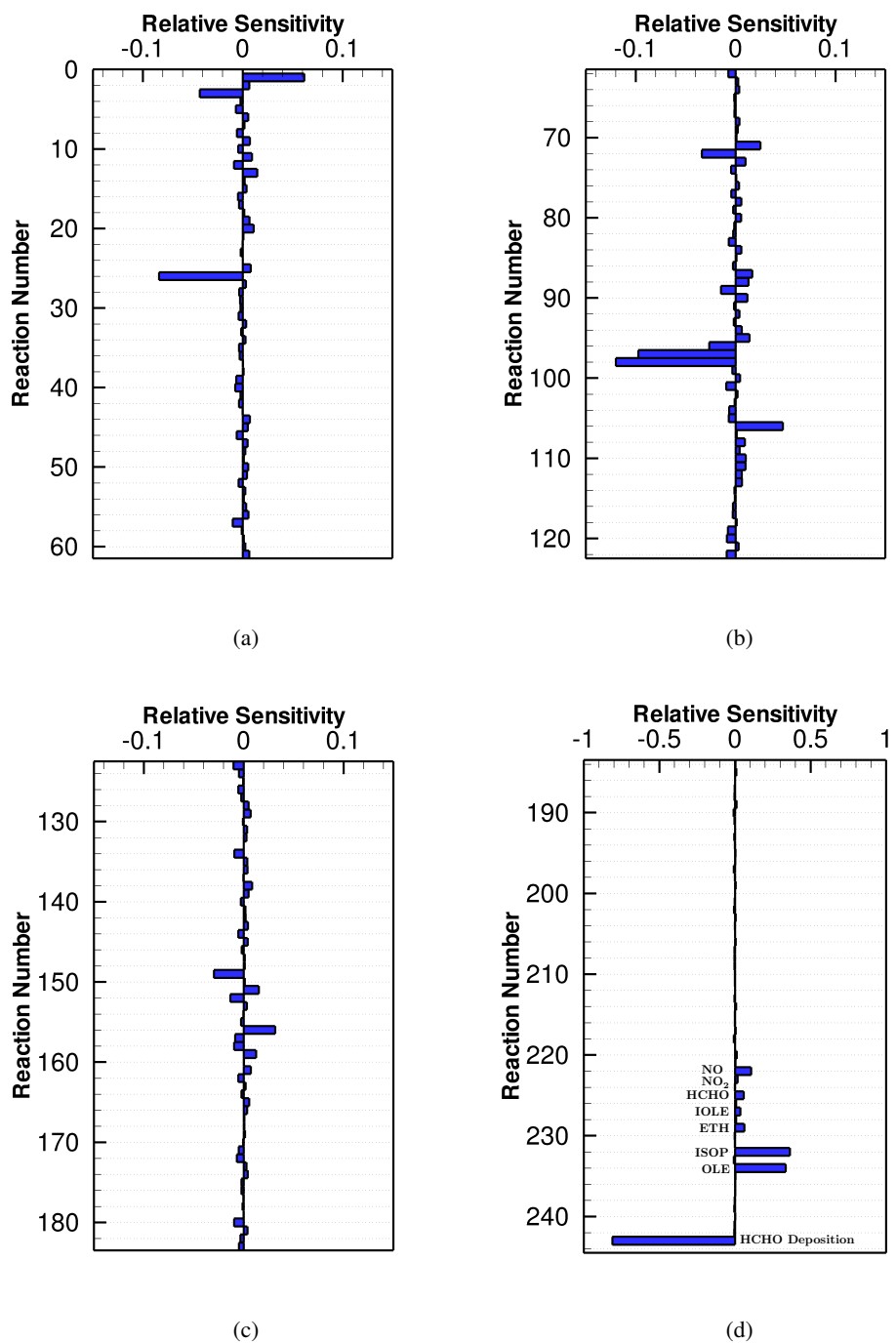

**Figure 8.** Averaged HCHO sensitivity to the CB6r3 mechanism over the 7-th day, when the surface emission is strong. All the values of the sensitivities shown in this figure can be found in Tab. S1 of the supplementary material.