# Peer review of "Study of Different Carbon Bond 6 (CB6) Mechanisms by Using a Concentration Sensitivity Analysis"

_Atmospheric Chemistry and Physics, 2020_

## Author Comment (AC1)

**Response to Referee #1**

The authors wish to thank Reviewer #1 for the comments, which greatly contribute to an improvement of the paper.

In the following, we address the issues raised by the reviewer:

**Q1.1:** Lines 217-219, model spin-up is an adjustment process that the model that moves from an initial state of unusual conditions to an equilibrium state. It is usually applied under conditions with surface emissions. The authors' claim here is thus inappropriate. I suggest removing these sentences.
**A1.1:** Thanks a lot for the advice. The reviewer is correct saying that the description about the spin-up process here is inappropriate. We have removed these sentences in the revised manuscript. Please see the corrections in lines 240-243 of the revised manuscript.

**Q1.2:** Line 298, I feel a little confused here. In a previous context (c.f. line 266), the authors already stated that (R158) is responsible for the discrepancy of the results between CB6r2 and CB6r3. But here they stated that (R158) is "possibly" the major reason, which is confusing. Please rephrase the sentence here.
**A1.2:** Yes, the description here is inappropriate. However, in the revised manuscript, this part has been deleted due to the update of the computational results. Still, many thanks for pointing it out.

**Q1.3:** Line 303, (R26) is not an HNO3 related reaction. Instead, it is an important $NO_3$ radical forming reaction, which plays an important role in the nighttime polluted atmosphere. Please rephrase it here.
**A1.3:** Thanks. We rephrased the sentences here. Please see lines 328-330 in page 11 of the revised manuscript.

**Q1.4:** Line 339, the estimation of ozone by CB6r3 would also be largely different from CB6r3 and CB6r2, isn't it? Please clarify it here.
**A1.4:** According to the results of the sensitivity analysis, under a different temperature condition, ozone predicted by CB6r3 would be substantially different from those predicted by CB6r1 and CB6r2, which is similar to the conclusion obtained from the sensitivity analysis of $NO_x$. We thus added more explanation here for clarify; please see lines 411-414 in page 13 of the revised manuscript.

**Q1.5:** Line 398, there is a redundant "of" in the sentence.
**A1.5:** Fixed. Thanks.

**Q1.6:** Line 413, from the response of ozone to the change of emissions in their study, it seems that the scenarios the authors investigated are in VOC-excess conditions so

that the increase of VOC tends to decrease ozone. I thus doubt that whether their conclusions are also valid under NO$_x$-excess conditions or not. The authors should provide a discussion (at least a brief one) on the limitations of their conclusions obtained in this paper.

**A1.6:** We agree with the opinion of the reviewer that the conclusions achieved in this manuscript are mostly valid under conditions that are focused on in this box model study. Whether these conclusions are still valid under different conditions or not needs further investigations. Therefore, according to the suggestion of the reviewer, we added a context stating the limitations of this study. Please see lines 660-662 in page 21 of the revised manuscript.

**Q1.7:** Code and data availability. It is always better to upload the source code as well as the data of the results to some website so that they can be shared with the scientific community. Only stating "the data can be acquired upon request" here is not enough from my point of view.

**A1.7:** Thanks. According to the reviewer's suggestion, we uploaded our files including the source code of the model and the data for the computational results to a webpage, so that the readers can download and share them. Please see the "Code and data availability" section in page 21 of the revised manuscript for the address of the webpage and the password to download these files.

**Q1.8:** The appendix table in the manuscript is misplaced. Please modify it.

**A1.8:** We have adjusted the location of the table in the revised manuscript. Thanks.

**Q1.9:** The manuscript is reasonably well written, but I would like to remind the authors that there are still a few sentences that can be improved significantly.

**A1.9:** We revised our manuscript again and corrected many inappropriate statements in the paper. Please see the words and sentences marked in red throughout the revised manuscript.

---

## Author Comment (AC2)

**Response to Referee #2**

The authors sincerely thank Reviewer #2 for the valuable comments and the very helpful considerations, which greatly contribute to an improvement of our paper.

In the following, we address the particular issues raised by Reviewer #2:

**Specific Comments**

**Q2.1:** The 7-day scenario with no emission has little relevance to how chemical mechanisms are used in atmospheric models. It is difficult to think of atmospheric conditions where an air parcel begins with substantial concentrations of ozone and precursors (initial $NO_x$ = 70 ppb with $VOC/NO_x$ = 4.8 and initial $O_3$ = 100 ppb, from Table 1) but then receives no input of either biogenic or anthropogenic emission over 7 days. It would be more atmospherically relevant to analyze the no emission simulations on day 2 or day 3. Figure 1 shows that the no emission simulations diverge by day 3 and, most likely, a main cause of differences after 2-3 days is the updated organic nitrate reactions of CB6r2 as summarized at line 110.

**A2.1:** Thanks a lot for the suggestion from the reviewer. In the no-emission scenario presented in the original manuscript, we switched the emissions off so that the influence caused by the changes in the chemical reactions between different CB6 mechanisms can be revealed. Otherwise, this influence may possibly be masked by the effect brought by the surface emissions.

However, we strongly agree with the reviewer's opinion that the conclusions achieved in the present study could be more meaningful under conditions that are more atmospherically relevant. Thus, we made the following improvements to the model and the set-up of the simulation scenarios as follows:

1. We replaced the no-emission case with a scenario implementing a weak emission, namely the "weak-emission" scenario. The other scenario with a larger emission intensity is thus called the "strong-emission" scenario in the revised manuscript. Similar to the strong-emission scenario, the level of hourly emissions for the weak-emission scenario was also taken from Saylor and Ford (1995) and Sandu et al. (1997), denoting a rural condition. Thus, this scenario can be interpreted as a situation that a polluted air parcel is transported to a rural region, in which the environment is relatively clean, so that the air parcel receives a small amount of emissions over a week.

2. We also refined our model by including dry deposition of atmospheric constituents. The corresponding description of how the dry deposition process is parameterized in the model can be found in lines 202-206 of the revised manuscript.

3. We also changed the initial air composition used in the two simulation scenarios under investigation in the present study, to make sure that the simulated levels of atmospheric constituents such as ozone and HCHO are in a reasonable range. For

this purpose, we discussed with Prof. Rolf Sander from Max-Planck Institute in Germany, who is in charge of the development of KPP and MECCA models and the compilation of NASA/JPL database. He suggested us to adjust the initial input of the model so that the outputs of the model reside in a reasonable value range that is normally detected in observations. Please see Tab. 1 in the revised manuscript for the modifications in the initial air composition, which are marked in red.

After the modifications of the model and the settings of the simulation scenarios, the mixing ratios of many atmospheric constituents such as ozone and HCHO obtained in the present study are currently more reasonable for atmospheric relevant conditions. For example, the simulated ozone in the weak-emission scenario ranges from 20 to 40 ppb over the last day of the simulation, while in the strong-emission scenario, it ranges from 60 to 130 ppb.

Of course, after all these modifications, the simulation results and the related discussions presented in the original manuscript need updated. Fortunately, many conclusions previously obtained still hold for the newly obtained results. We have updated all the simulation results and the related discussions in the revised manuscript, which are marked in red.

The reviewer suggested us to analyze the results at an early stage of the computation. However, we feel that at that time, the reaction system has not reached the state of chemical equilibrium, and is mostly affected by the initial air composition used in the scenario. Thus, in our opinion, analyzing these results may not be complete and appropriate. Instead, we refined our model and the set-up of the simulation scenarios as described above, and then analyzed the updated model results averaged over the last day of the computation like before. Still, many thanks for the suggestion of the reviewer.

**Q2.2:** The 7-day scenario with emission produces O3 above 300 ppb which is rarely observed in the atmosphere at ground level. In the atmosphere, O3 accumulation is moderated by surface deposition and diluted by daily increase and decrease in the boundary layer depth. It is unclear whether the box model included either deposition or dilution due to cycling of boundary layer depth. If the effects of deposition and dilution cannot be included in the box model, the model results can be analyzed on day 2 or day 3 when the O3 concentration is closer to atmospherically relevant conditions.

**A2.2:** This question is similar to the question above, **Q2.1**, saying that the computational result of the model is out of a normal value range in observations. As discussed above, after the refinements of the model and the set-up of the simulation scenarios, at present, the ozone level obtained in the "strong emission" scenario after the 7-day simulation resides in a range of 60-130 ppb, which is normally observed for the surface ozone in field campaigns. The following sensitivity analysis and the discussions are thus more meaningful for atmospheric relevant conditions. Please see the context in Section 3.3 and Section 3.4 marked in red for the updated results and

the discussions.

The reviewer also mentioned the role of deposition and dilution in the model. As mentioned in **A2.1**, we currently include the dry deposition process in the model (see lines 203-207 in the revised manuscript). However, the dilution caused by the change of the boundary layer height has not been considered at present. Instead, we used a daily-averaged boundary layer height, 1km, in the model, as we wish to focus more on the influence on the change of atmospheric constituents brought about by the change of chemical reactions between different mechanisms instead of other physical processes that will add more complexity to the analysis of the computational results.

**Q2.3:** The HCHO concentration in simulations with emission reaches 50 ppb which is high compared to atmospheric measurements. HCHO may accumulate too much because the box model has no dilution due to cycling of the boundary layer depth. Conducting the sensitivity analysis with very high HCHO concentrations limits relevance to the atmosphere. The discussion at lines 467 to 487 compares the box model HCHO sensitivity results to results of 3D simulations and ambient measurements but it is difficult to have confidence in these comparisons because the box model HCHO concentrations are about an order of magnitude larger than in the 3D simulations.
**A2.3:** Thanks. After the improvements of the model, the HCHO levels obtained in the present study for the weak-emission scenario and the strong-emission scenario reside in a value range of 0.5-1.5 ppb and 7-15 ppb, respectively. In previous observations of HCHO at the urban sites, the surface value of HCHO ranges from 2.5 ppb to 27 ppb (Ling et al., 2017). Thus, after the improvements of the model, at present, the estimated level of HCHO in this study is more reasonable compared with the level in observations, which makes the following sensitivity analysis more reliable and meaningful for atmospheric relevant conditions. Please see the context marked in red in lines 448-458 of the revised manuscript.

**Q2.4:** The study conclusions are limited to the box model conditions that have been analyzed and presented. At minimum, the discussion of conclusions should clearly state that conclusions are drawn from box model conditions that are substantially different from the conditions present in most 3D model simulations of the atmosphere. A better solution would be to reanalyze the box model results focusing on day 2 or 3 and then update the conclusions, which may add new conclusions. The same comment applies to the summary of conclusions included in the abstract.
**A2.4:** As mentioned above, the present model after the improvements now gives reasonable levels of atmospheric constituents such as ozone and HCHO, which are not far from the conditions present in 3-D models. However, according to the suggestion of the reviewer, we added a short description about the limitations of the present study in the revised manuscript. In this description, we clearly state that the obtained conclusions in the present study are mostly valid under conditions that are investigated in this box model; please see lines 660-662 in the revised manuscript.

Thanks a lot for the suggestion.

**Q2.5:** Figure 2 and similar Figures present much information in a concise format and are useful. However, identifying the contribution of an individual reaction in Figure 2 is difficult. The authors can add a Table in the supplementary material listing the values of S(ij) from Figure 2 and the similar Figures.

**A2.5:** According to the suggestion of the reviewer, we added three tables in the revised supplementary material (i.e. Tabs. S1-S3), listing all the values of the sensitivities belonging to the two simulation scenarios presented in this study. Please see Tabs. S1-S3 in the revised supplementary material.

**Q2.6:** The manuscript is written clearly although one aspect of language should be clarified. The word "discrepancy" means a difference that is not expected. In this study, the mechanism versions are different and so they produce different concentrations which isn't a discrepancy. Change the word discrepancy to difference.

**A2.6:** We changed the word "discrepancy" to "difference" or "deviation" throughout the manuscript. Thanks a lot for the correction.

**Technical Corrections**

**Q2.7:** Line 8: Using the model species name CXO3 in the abstract will be difficult for many readers to understand. This statement could be re-written to say that the fate of larger PAN-type compounds (PANX) is influential in these box model scenarios.

**A2.7:** We added the description of CXO3 in the abstract of the revised manuscript. Please see lines 7-8 in the abstract. For the context about PANX, it has been deleted in the revised manuscript due to the change in the computational results.

**Q2.8:** Line 20: Harmful effects of air pollution occur at concentrations lower than air quality standards.

**A2.8:** We have rephrased the sentence. Please see line 23 of the revised manuscript.

**Q2.9:** Line 37: The correct citation for the Carbon Bond lumping method is Gery et al., 1989

**A2.9:** Corrected, thanks.

**Q2.10:** Line 74: This paragraph doesn't mention previous relevant studies (Derwent 2017, 2020) that compare CB6 to other mechanisms. It would be useful to cite these studies and note that Derwent used CB6r3 although with changes to the inorganic reactions so that all the mechanisms compared had the same inorganic reactions.

**A2.10:** We cited the references suggested by the reviewer and added the relevant discussions. Please see lines 90-97 in the revised manuscript.

**Q2.11:** Line 86: The statement that changes brought about by mechanism updates are "unknown" is contradicted by the review of earlier findings presented in the preceding paragraph. A different statement that could be made is that the motivation of this study is to better understand the effects of mechanism changes. The same comment applies at line 3.

**A2.11:** The reviewer is correct saying that the statement here is inappropriate. We have rephrased the corresponding sentences. Please see lines 99-100 in page 4 and lines 2-3 in the abstract.

**Q2.12:** Table A1 includes species that are part of the CMAQ aerosol scheme (i.e., TERPRXN, BENZRO2, TOLRO2, XYLRO2, PAHRO2) but play no part in CB6 gas-phase chemistry (i.e., they have no gas-phase removal reactions). It would be clearer to delete them from Table A1. Also, CMAQ changed the name of the CB species representing xylenes (XYL) to XYLMN (meaning xylene minus naphthalene) and added a naphthalene species (NAPH) for SOA chemistry.

**A2.12:** We have deleted the species TERPRXN, BENZRO2, TOLRO2, XYLRO2, PAHRO2 in Tab. A1 for clarify as the reviewer suggested. Moreover, we changed the name "XYLMN" to "XYL". Thanks a lot for pointing it out.

**Q2.13:** Table A1. Add a Table footnote that reactions 223 to 231 were added to the mechanisms in this study to represent box model emissions but they aren't part of the CB6 mechanisms.

**A2.13:** Done, thanks. Please see the footnotes of Tab. A1 in the revised manuscript.

**Q2.14:** Table A1. For clarity, add a Table footnote to the table specifying where the authors obtained each version of CB6. This Table is cleverly constructed.

**A2.14:** Done, thanks. Please also see the footnotes of Tab. A1 in the revised manuscript.

**References:**

Sandu, A., Verwer, J., Loon, M. V., Carmichael, G., and Seinfeld, J.: Benchmarking stiff ODE solvers for atmospheric chemistry problems I: implicit versus explicit, Atmos. Environ., 31, 3151–3166, 1997.

Saylor, R. D. and Ford, G. D.: On the comparison of numerical methods for the integration of kinetic equations in atmospheric chemistry and transport models, Atmospheric Environment, 29, 2585-2593, 1995.

Ling, Z., H., Zhao, J., Fan, X.M., Wang, X.M.: Sources of formaldehyde and their contributions to photochemical $O_3$ formation at an urban site in the Pearl River Delta, southern china. Chemosphere, 168, 1293-1301, 2017.

---

## Author Response (AR2)

**Response to Reviewer#2**

Thanks a lot for the comments of the reviewer. We made the following modifications in the revised manuscript.

**Q2.1:** Thanks for carefully responding to review comments including adding Tables S2 and S3 in the supplement. However, the number format in these Tables shows many sensitivities as 0.00 which isn't useful. Please use an exponential format like 0.00E+00 so that the full range of sensitivities can be understood and compared.

**A2.1:** We changed the format of the data shown in Tabs. S1-S3 of the supplementary material, according to the suggestion of the reviewer. Thanks.

**Q2.2:** In the context of the revised manuscript the citation to ENVIRON (2015) is outdated and can be updated to Ramboll (2020). Ramboll: User's Guide: Comprehensive Air Quality Model with Extensions (CAMx), Version 7.10, Tech. rep., 2020. Available at https://www.camx.com/about/

**A2.2:** We added the citation suggested by the reviewer. Thanks for the information.